# Visual Explanations for Capsule Networks

**Saja Tawalbeh**                                                          *saja.tawalbeh@uantwerpen.be*
*University of Antwerp, sqIRL/IDLab, imec*

**José Oramas**                                                            *jose.oramas@uantwerpen.be*
*University of Antwerp, sqIRL/IDLab, imec*

**Reviewed on OpenReview:** *https: // openreview. net/ forum? id= eNQK9WJkid*

## Abstract

The limited availability of explainability methods for Capsule Networks (*CapsNets*) restricts their adoption in several domains, including vision and other structured data. Although *CapsNets* offer structured and interpretable representations, existing explanation methods have primarily focused on more traditional Convolutional Neural Networks (*CNNs*) and are not directly applicable to capsule-based architectures. To address this issue, we propose an explanation method (*Caps-CAM*), which generates attribution maps to justify the predictions made by feed-forward *CapsNet* architectures. Unlike prior explanation methods for *CapsNets* that adapt techniques originally designed for *CNNs*, *Caps-CAM* explicitly employs gradient information that reflects the relevance of each capsule to a class of interest. As the gradient can help highlight the most relevant capsules, each selected capsule activation map is weighted by its corresponding gradient. The final attribution heatmap is then generated as a linear combination of weighted activation maps based on their contribution to the target class. Empirical comparisons w.r.t. state-of-the-art explanation techniques previously introduced for *CNNs*, show that *Caps-CAM* can effectively serve as an explanation method for *CapsNets*. Experiments on standard and real-application data sets show the effectiveness of the introduced *Caps-CAM*.

## 1 Introduction

Explainability for Deep Neural Networks (*DNN*) aims to enhance transparency by exposing aspects of the inference process that could be informative for its users. Most of these methods were originally designed to explain Convolutional Neural Networks (*CNNs*). Popular methods tailored to this type of architecture include gradient-based methods (Simonyan et al., 2013) and class activation mapping (*CAM*) (Zhou et al., 2016). Gradient-based methods propagate the gradient of a given class in order to emphasize the image regions that have the most significant impact on the prediction of that class. In contrast, *CAM* methods leverage the activation maps from convolutional layers for this purpose, with some variants that also exploit gradient information.

Capsule Networks (*CapsNets*) (Sabour et al., 2017; Hinton et al., 2018) have been introduced as an extension to standard neural networks. Unlike traditional *DNNs*, *CapsNets* are designed to explicitly model hierarchical relationships in the data, which inherently enhances their interpretability capabilities. The growing interest in *CapsNets* has led to their application in various domains, including medical imaging (Ramana et al., 2022; Ayidzoe et al., 2022; Geng et al., 2025; Srinivasan et al., 2025; Sushith et al., 2025), NLP (Wang et al., 2020b; Chen et al., 2024; Liu et al., 2024), and computer vision (Renzulli et al., 2023; Everett et al., 2024; Zeng et al., 2025). This has resulted in an increasing need to explain the decisions made by such networks.

While much work has focused on designing interpretability methods to understand what a capsule-based model has learned (Li et al., 2025; Geng et al., 2025; Tawalbeh & Oramas, 2024; Mitterreiter et al., 2023; Bondarenko et al., 2023; Gu et al., 2021), these methods primarily focus on analyzing internal representations

such as capsule activations or *part-whole* relationships. In contrast, Everett et al. (2024); Konstantinou et al. (2025) focus on improving self-supervised equivariant representation learning and use capsule geometry as a functional component for learning structured representations. Rather than explaining *CapsNet* predictions.owever, understanding what the model has learned does not necessarily provide insight into why a specific decision was made for a given input. This gap highlights the need for explanation methods that can attribute model predictions to specific inputs. Despite this, relatively fewer efforts have specifically focused on explaining the predictions made by *CapsNets* (Nguyen et al., 2019; Shen & Gao, 2018). Efforts to explain the predictions made by these networks (Nguyen et al., 2019; Ren et al., 2019) have primarily exploited the routing algorithms that characterize capsule architectures. While these methods offer certain advantages, their reliance on the specifics of the routing mechanisms (i.e., dynamic routing) limits their applicability to other architectures based on different routing methods. We argue that eliminating this dependency is crucial for developing explanation methods that are agnostic towards the underlying feed-forward architecture. Another trend in the visual explanation of *CapsNets* has been the adaptation of techniques originally designed for CNN-based models to capsule architectures (Shen & Gao, 2018; Ramana et al., 2022; Tawalbeh & Oramas, 2024). While effective to some extent, these CNN-tailored methods are designed to treat the output of such networks as a scalar, which is normal for standard *CNNs*. Consequently, they ignore key characteristics of *CapsNets* and fail to exploit the (*part-whole*) mechanisms by which *CapsNets* operate.

Building on the above observations and inspired by Wang et al. (2020a), **our main contributions** are firstly, proposing *Caps-CAM*, a visual explanation method for feed-forward *CapsNet* architectures. The proposed method takes a different approach from *CNNs* based methods by first leveraging gradients to measure the relevance of the capsules on the decision-making process and retaining only the most relevant ones after ranking them. Once the top relevant capsules have been identified, we compute their contribution by multiplying the activation maps and the corresponding top relevant capsules. By selecting only the top relevant capsules prior to computing their contributions, this two-step process eliminates the dependence on specific routing algorithms. Consequently, the proposed method is compatible with different routing and structural *CapsNet* designs, which allows it to be applied across architectures with varying routing mechanisms and structural designs. Moreover, by restricting computations to the most relevant capsules rather than processing all capsules, it avoids redundant computations that would otherwise be required. To this end, this work introduces *Caps-CAM* as a post-hoc explainability framework for justifying the predictions of existing feed-forward *CapsNets*. The proposed method operates independently of architectural modifications, routing changes, or retraining procedures.

For evaluation, we consider several existing methods such as the work of Selvaraju et al. (2017); Chattopadhay et al. (2018); Wang et al. (2020a), showing that the proposed *Caps-CAM* achieves competitive quantitative performance (in terms of Average % Drop and *AOPC*). On the qualitative side, our explanations show that the considered *CapsNet* architectures exploit both object-specific and context-related features when making predictions. Secondly, testing *Caps-CAM* on a more realistic setting (i.e. the MSTAR/application dataset), demonstrates the practical value of the proposed method. Overall, the experiments conducted on both standard benchmarks and a real-world application dataset demonstrate the capabilities of the proposed *Caps-CAM* method in generating meaningful visual explanations.

## 2 Related Work

This section focuses on methods that have been proposed for explaining the predictions made by *CapsNets* through a post-hoc analysis framework. Therefore, the limitations discussed in this section are specifically related to the limited availability of explanation methods designed for *CapsNets* in the existing literature, which motivates the need for the proposed method.

*CapsNets* (Sabour et al., 2017) were specifically designed for modeling *part-whole* relationships between features in the data used to train them. Several efforts exploited this characteristic, primarily through the routing mechanism, to explain the predictions made by these networks. In line with this, Shahroudnejad et al. (2018) identified the relevant activation paths within the network and used the most relevant units along these paths as means for explainability. Jia & Huang (2020) extended the methodology from Phaye et al. (2018), focusing on explaining global features utilizing a set of relevant capsules. These capsules are selected

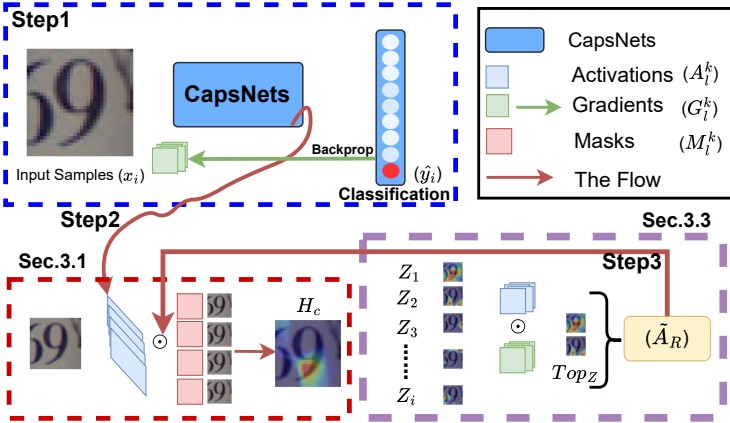

Figure 1: Overview of *Caps-CAM* methodology: Step 1 involves a forward pass through the model. In Step 2, $A_l^k$ is extracted for use in *Score-CAM*. Finally, Step 3 employs $A_l^{k,\intercal}$ and $G_l^{k,\intercal}$ to identify and select the top capsules of *Caps-CAM*. The section refers to where these steps are introduced in the paper.

using routing algorithms that establish connections between the intermediate layers and the later/class capsule. Different from the methods above, which provide explanations by employing the routing to verify the relevant activation paths, the following methods employ the routing paths/mechanism to generate attribution heatmaps. For instance, Tawalbeh & Oramas (2024) produces gradient-based attribution heatmaps to justify the prediction by highlighting the contribution of specific units in a given layer to a given class. In their method, the selection of these units is determined by the routing algorithm, which makes the resulting attribution maps inherently dependent on this algorithm. Similarly, in Ren et al. (2019), the visual heatmaps are produced by the combination of class-relevant units modulated by their associated coefficients in the routing mechanism of the architecture. In contrast to Ren et al. (2019) and Tawalbeh & Oramas (2024), our proposed method eliminates the dependence on such routing algorithms. Instead, we rely on internal activation maps in conjunction with capsule-related gradients to address the explanation task. Such methods rely explicitly on routing coefficients and the internal units as a means of explanation. For instance, the work of Shahroudnejad et al. (2018) the routing coefficients computed from the activations of a given input image as a means of explanations, which is a completely different methodology w.r.t our proposed method.

A second group of efforts has focused on the adaptation of methods, initially designed for *CNNs*, for the explanation of *CapsNets*. Along this line, gradient-based methods have been explored in the work of Tawalbeh & Oramas (2024), while guided backpropagation was applied in Phaye et al. (2018). Additionally, *Grad-CAM* has been adapted to produce visual explanations for capsule-based architectures (Shen & Gao, 2018; Mobiny et al., 2019). Unlike routing-based methods, these methods do not depend on a specific routing procedure and can therefore be evaluated across the same architectures as our method. On the one hand, the work of Tawalbeh & Oramas (2024) focuses on the visual inspection of specific units from intermediate layers. While this provides insight into the contribution of that specific unit, it does not account for the collective contribution of the entire network. As a result, it is not directly comparable to our proposed method, which considers the entire architecture. On the other hand, Grad-CAM is used by Shen & Gao (2018); included as a baseline against which we compare our method. Therefore, an empirical comparison with the most relevant prior CapsNet-specific explanation methods cannot be included in this paper, as these methods produce fundamentally different types of explanations (e.g., non-heatmap outputs), making them not directly comparable to the proposed approach.

Different from the methods mentioned above and similar to *Score-CAM* (Wang et al., 2020a), our proposed method utilizes the relevance of each internal activation channel alongside their corresponding gradients w.r.t the predicted class by weighting them based on the class scores. This produces an explanation heatmap that allows the identification of regions of the input that determine the predicted class.

# 3 Proposed Method

This section introduces the *Caps-CAM* method for explaining predictions made by *CapsNets* without altering their original structure or inference process. This enables the generation of visual explanations while preserving the native behavior and performance of the underlying CapsNets. We first provide background details on *Score-CAM* (Wang et al., 2020a), as our methodology builds upon it. Then, we cover different aspects of the proposed method. To summarize, the proposed method utilizes activation maps while incorporating gradients into the proposed framework in order to compute relevance matrices. Then, the matrices from the most relevant capsules are used to produce explanations in the form of visual heatmaps. Fig. 1 illustrates the pipeline of the proposed *Caps-CAM* explanation method.

## 3.1 Score-Weighted Class Activation Mapping

Methods based on Class Activation Mapping (CAM) (Zhou et al., 2016; Selvaraju et al., 2017) produce explanation heatmaps through the weighted sum of the activations of the filters in the last convolutional layer of a *CNN* architecture. A common practice is to compute these weights from gradients related to the predicted class. In contrast, *Score-CAM* (Wang et al., 2020a) removes the requirement on gradient information by relying solely on forward passes and internal activations.

Formally, let $F(\cdot)$ denote a pretrained *CapsNet*, which can be decomposed into a backbone feature extractor $b(\cdot)$ and a classification head. Given an input $x_i$, the network produces class scores $S_i = F(x_i)$, from which the predicted class label $\hat{y}_i$ is obtained via $\hat{y}_i = \arg\max_c S_i(c)$. Moreover, let $A_l^k$ denote the activation of the $k$-th capsule (or feature channel) at layer $l$. Hence, the activations $A_l^k(x_i)$ for a given input $x_i$, obtained during the forward pass, are extracted from the backbone representation $A_i = b(x_i)$ as described in Eq. 1.

$$A_l^k = b_l^k(x_i), \quad \forall k \tag{1}$$

To avoid ambiguity, we emphasize that the *Caps-CAM* pipeline, based on *Score-CAM*, includes intermediate tensor reshaping and alignment steps between activation extraction and capsule operations. These transformations are applied before any relevance computation and are summarized in Algorithm 1.

For *Score-CAM*, the activation maps extracted from the layer prior to the classification part are upsampled to correspond with the size of the input $x_i$. We denote the resulting upsampled activation map as $\hat{A}_l^k(x_i)$. These upsampled activation values $\hat{A}_l^k(x_i)$ are then scaled to the range $[0, 1]$ via min-max normalization. This is followed by the occlusion phase, where the input $x_i$ is perturbed using each of the upsampled activation maps $\hat{A}_l^k(x_i)$, producing a new masked input $M_l^k$ (Eq. 2).

$$M_l^k = \hat{A}_l^k(x_i) \odot x_i \tag{2}$$

where $\odot$ represents the element-wise multiplication between the upscaled activation map $\hat{A}_l^k(x_i)$ and the input $x_i$. Then, the importance of the $k^{th}$ activation map is determined by performing a forward pass through the pretrained network $F$ with the masked input $M_l^k$ and extracting only the output score corresponding to the target class predicted for the original (unmasked) input, defined as $\tilde{S}^k = F(M_l^k)$. The final step involves

| Capsule Network | MNIST | SVHN | CIFAR-10 | MSTAR |
|:---:|:---:|:---:|:---:|:---:|
| DR | 99.1 | 91.0 | 74.2 | 99.4 |
| DR-Conv | 99.0 | 90.5 | 74.0 | 99.7 |
| DR-Tra | 99.4 | 91.0 | 76.0 | 99.3 |
| EM | 98.0 | 80.0 | 72.2 | 86.6 |

Table 1: *CapsNets* test accuracy (%) on MNIST, SVHN, CIFAR-10, and MSTAR datasets. Dynamic Routing (DR), Dynamic Routing with deep convolutional layers (*DR-Conv*), Dynamic Routing with transformer blocks before the *PC* layer (*DR-Tra*), and Expectation Maximization routing based network (*EM*).

applying a *ReLU* activation function to the combination of the feature maps $\hat{A}_l^k$ using the weights $\tilde{S}^k$ (Eq. 3). This is done to focus on the features that have a positive contribution to the predicted class $c$.

$$H_c = ReLU(\sum_{k=1}^{k} \tilde{S}^k \hat{A}_l^k) \tag{3}$$

## 3.2 Class Activation Mapping for *CapsNets*

This section focuses on adapting *Score-CAM* to capsule-based architectures, which serves as the foundation for the proposed method that will be introduced in the following section (Sec. 3.3). In contrast to *CNN* models, *CapsNets* are composed of so-called capsule elements that fundamentally characterize these architectures. While adopting CNN-based methods is possible, doing so poses challenges for directly applying the *Score-CAM* method to these architectures. This difficulty arises from the potential loss of encoded information within each capsule dimension. To address this, additional steps should be taken immediately after performing the forward pass mentioned in Eq. 1. Specifically, the activations $A_l^k \in \mathbb{R}^{1 \times \mathbb{Z}}$ (with $\mathbb{Z}$ being the dimensionality of the activations output) are transposed to rearrange their dimensions, yielding $A_l^{k,\intercal}(x_i) \in \mathbb{R}^{1 \times Z \times K \times H \times W}$, to incorporate the capsule dimension $Z$, which represents the capsule elements.

Afterwards, a summation is performed over the capsule dimension $Z$ of the reshaped activations $A_l^{k,\intercal}(x_i)$, resulting in the aggregated tensor across the capsule dimension $Z$ into a single unified matrix $\tilde{A}_l^k(x_i) \in \mathbb{R}^{1 \times K \times H \times W}$. By doing so, this aggregation simplifies the representation, reduces the dimensionality of the capsules, and fuses the information they encode. It is important to note that, unlike standard *CAM* methods, where no input masking is applied, similar to *Score-CAM*, the proposed method incorporates a masking step after normalization to adapt the process to capsule-based architectures. Specifically, instead of normalizing the activation maps independently, we normalize them by considering the global min-max values across all matrices $[A_l^{k\intercal}(x_i)_1, A_l^{k\intercal}(x_i)_2; ...; A_l^{k\intercal}(x_i)_z]$ associated with the capsule dimension $Z$. The normalization is performed prior to summing across the capsule dimension and subsequently masking the input using $\tilde{A}_l^k(x_i)$. By doing so, it ensures that all units are calibrated and that no particular unit dominates the aggregated activation map, providing a comparable contribution from the capsule dimensions.

## 3.3 *Caps-CAM*

The formulation introduced in Sec. 3.2 establishes a capsule-aware adaptation of *Score-CAM* by restructuring and normalizing capsule activations. Building upon this foundation, *Caps-CAM* extends the same pipeline by introducing a gradient-guided method to estimate capsule relevance, followed by a top-capsule selection. This extension allows the method to focus on the most informative capsule dimensions, which improves the quality of the resulting visual explanations.

The proposed method started by modifying the original *Score-CAM* immediately after extracting the activations $A_l^k$, prior to the masking step (defined in the previous section). In this step, $A_l^k$ are weighted using the corresponding gradients to identify the relevant capsules, ensuring that only the most important ones are retained for subsequent processing. Towards this goal, we first reintroduce the use of gradients. This is motivated by the ability of gradients to capture crucial information regarding units that determine the predicted class.

To achieve this, we compute the gradients $G_l^k(x_i, \hat{y}_i) = \frac{\partial S_i(\hat{y}_i)}{\partial A_l^k(x_i)}$. We reshape the activations $A_l^{k,\intercal}(x_i)$, which were previously introduced and preprocessed in Sec. 3.2. Simultaneously, we obtain the gradients $G_l^k(x_i, \hat{y}_i)$ for the predicted class $\hat{y}_i$, computed from the score $S_i(\hat{y}_i)$ with respect to the same input $x_i$. Then, we apply the same transpose to gradients, producing the transposed matrix $G_l^{k,\intercal}(x_i, \hat{y}_i)$, and ensure that both matrices, $G_l^{k,\intercal}(x_i, \hat{y}_i)$ and $A_l^{k,\intercal}(x_i) \in \mathbb{R}^{1 \times Z \times K \times H \times W}$, have the correct shape, which guarantees their compatibility for the subsequent steps.

With $G_l^{k,\intercal}(x_i, \hat{y}_i)$ and $A_l^{k,\intercal}(x_i)$ in place, we proceed to compute the most influential (relevance) capsules at the dimension $Z$ level in the following step. Specifically, this is achieved by computing the product between

the dimension $Z$ of a given capsule within $G_l^{k,\mathsf{T}}(x_i, \hat{y}_i)$ and the corresponding dimensions $Z$ within $A_l^{k,\mathsf{T}}$ of the same capsule. This operation indicates how strongly each dimension contributes to the loss, while the activations reflect the magnitude of the response of each dimension. The key insight here is that the relevance of each capsule is determined based on its contribution to the final prediction by employing both matrices. This enables us to identify the relevance score of each capsule $\mathcal{R}_i^k$ as calculated in Eq. 4. The resulting tensor $\mathcal{R}_i^k$ represents the relevance of each capsule, where the activations are weighted by the magnitude of the gradients with respect to the target class score. The gradient magnitude captures the sensitivity of the model output to each capsule, which provides a measure of its contribution to the final decision.

$$\mathcal{R}_i^k = A_l^{k,\mathsf{T}}(x_i) \cdot G_l^{k,\mathsf{T}}(x_i, \hat{y}_i), \quad \forall k, \quad \mathcal{R}_i^k \in \mathbb{R}^{1 \times Z \times 1 \times 1 \times 1} \tag{4}$$

To complement the previous step, we compute the product between the activations $A_l^{k,\mathsf{T}}$ and their corresponding relevant capsule scores $\mathcal{R}_i^k$. This ensures that the contributions of the most relevant capsules in the dimension $Z$ are emphasized over the rest. Therefore, the relevance activation matrix $\mathbf{A}_R^k$ is calculated as:

$$\mathbf{A}_R^k = A_l^{k,\mathsf{T}}(x_i) \cdot \mathcal{R}_i^k, \quad \forall k, \quad \mathbf{A}_R^k \in \mathbb{R}^{1 \times Z \times K \times H \times W} \tag{5}$$

By computing the relevance activation matrix $\mathbf{A}_R^k \in \mathbb{R}^{1 \times Z \times K \times H \times W}$ this way, we eliminate the dependence on routing algorithms to select the relevant capsules. This design choice improves the compatibility of the proposed method with different capsule-based architectures, which enhances the overall applicability of our method. This is also noted in the study from Byerly et al. (2021), which suggests that the complexity introduced by routing algorithms in *CapsNets* can be avoided, as these networks can be effectively trained without relying on such mechanisms.

Then, the capsules are ranked in descending order of their magnitude scores, which encodes capsule relevance along the $Z$ dimension. The top-ranked capsules are subsequently selected for further processing. Based on the above, we select the top capsules based on their maximum magnitude, which are obtained from the relevance matrix $\mathbf{A}_R^k$. This results in a tensor of indices $\mathbf{I}_{top}$ corresponding to the most relevant capsules (the top capsules), which will be used in subsequent steps. Then, we modify $\mathbf{A}_R^k$ to focus the attention on the relevance of the selected $\mathbf{I}_{top=2}$ capsules, using the previously obtained top ranking ones. Hence, the final relevance activation matrix $\mathbf{A}_{R,Z}^*$ is obtained by retaining the indices corresponding to the $\mathbf{I}_{top=2}$ capsules (Eq. 6).

$$\mathbf{A}_{R,top}^* = \mathbf{A}_R^k[\mathbf{I}_{top}], \quad \mathbf{A}_{R,top}^* \in \mathbb{R}^{1 \times 2 \times K \times H \times W} \tag{6}$$

Having $\mathbf{A}_{R,top}^*$ appropriately dimensioned, a summation is performed across the capsule dimensions $Z$, resulting in $\tilde{\mathbf{A}}_R \in \mathbb{R}^{1 \times K \times H \times W}$. Once this summation is complete, we first apply a capsule-based normalization step (as explained earlier in Sec. 3.2) to $\tilde{\mathbf{A}}_R$, which ensures that their values are appropriately scaled. Second, we return to the original *Score-CAM* formulation and apply the remaining steps to the modified matrix $\tilde{\mathbf{A}}_R$ (similar to $A_l^k(x_i)$ in the original *Score-CAM* formulation). We then proceed with masking the input using $\tilde{\mathbf{A}}_R$ and combine it with the calculated scores. Retaining only the top capsules is crucial, as some capsules are strongly activated while others exhibit weak responses. The weak capsules should be removed from the computation to prevent their noisy contributions, which can lead to diffuse explanation heatmaps.

To clarify the role of gradients in Caps-CAM, gradients are not directly used to construct the final heatmap. Instead, they are only used to guide the selection of the most informative capsule dimensions associated with the predicted class, while the final explanation is generated from the corresponding activation maps using score-based weighting. During training, we also monitor the optimization and loss curves to avoid convergence to local minima or saturated regions. This design reduces the sensitivity issues typically associated with purely gradient-based methods, as the final heatmap is mainly driven by activation information rather than raw gradient values.

It is important to note that gradients are not assumed to be inherently stable or causally indicative of importance; instead, they are employed as a practical heuristic for estimating capsule relevance, whose effectiveness is validated empirically.

See Algorithm 1 for pseudocode listing the different steps involved in the proposed *Caps-CAM* method.

---

**Algorithm 1** Caps-CAM explanation method

---

1: **Input:** $x_i$ (input sample), $F$ (CapsNet), $l$ (target layer)
2: **Definitions:** $Z$ (capsule dimension), $k$ (filters), $\tau$ (top-$k$ capsules)
3: **Forward pass**
    $S_i \leftarrow F(x_i)$
    $\hat{y}_i \leftarrow \arg\max_c S_i(c)$
    Extract activations $A_l^k(x_i)$
4: **Gradient computation**
    Compute $G_l^k(x_i, \hat{y}_i)$
    Reshape $A_l^k,\ G_l^k \in \mathbb{R}^{1 \times \mathbb{Z}}$
    $A_l^{k,\intercal} \leftarrow A_l^k; A_l^{k,\intercal} \in \mathbb{R}^{1 \times Z \times K \times H \times W}$
    $G_l^{k,\intercal} \leftarrow G_l^k; G_l^{k,\intercal} \in \mathbb{R}^{1 \times Z \times K \times H \times W}$
    $\mathcal{R}_i \leftarrow A_l^{k,\intercal} \cdot G_l^{k,\intercal}; \mathcal{R}_i \in \mathbb{R}^{1 \times Z \times K \times H \times W}$
    $\mathbf{A}_R^k \leftarrow A_l^{k,\intercal} \cdot \mathcal{R}_i; \mathbf{A}_R^k \in \mathbb{R}^{1 \times Z \times K \times H \times W}$
5: **Top capsule selection**
    The magnitude is in $(A_R^k)$
    Rank magnitude $(A_R^k)$ in descending order
    $\mathbf{I}_{top} \leftarrow \text{index[top-2 capsules]}$
    $\mathbf{A}_{R,top}^* \leftarrow \mathbf{A}_R^k[\mathbf{I}_{top}]; \mathbf{A}_{R,top}^* \in \mathbb{R}^{1 \times 2 \times K \times H \times W}$
6: **Aggregation**
    $\tilde{\mathbf{A}}_R \leftarrow \sum_Z \mathbf{A}_{R,top}^*$
7: Apply capsule-based normalization to $\tilde{\mathbf{A}}_R$
8: $\mathbf{H}_{CapsCAM} \leftarrow \text{Score-CAM-Mask}(\tilde{\mathbf{A}}_R)$
9: **return** $\mathbf{H}_{CapsCAM}$

---

## 4 Evaluation

This section presents the evaluation procedure employed to validate the proposed method, along with the corresponding results. First, *Caps-CAM* is qualitatively evaluated through visualizations on several datasets. Second, the validity of the visual explanations is assessed, specifically focusing on the importance of the highlighted regions in influencing the outputs of the model.

**Implementation details.** Our experiments are conducted with NVIDIA GeForce GTX 1060 and DGX-2. The considered architectures are implemented in PyTorch and trained for 60 epochs and a batch size of 32. We use the adam optimizer with the default learning rate, dropout, and weight decay. For the routing algorithm, 3 routing iterations were used in all *CapsNet* architectures. During training, for the above-mentioned datasets, no data augmentation or hyperparameter tuning was applied. For the *EM* network architectures, the learning rate was set to $3 \times 10^{-3}$, allowing for effective convergence during training. We employed a weight decay of $2 \times 10^{-7}$ to prevent overfitting. The code is publicly available at [1].

**Datasets.** Following Sabour et al. (2017); Hinton et al. (2018); Gu et al. (2021); Byerly et al. (2021); Tawalbeh & Oramas (2024); Mitterreiter et al. (2023), we adopt well-established datasets for the evaluation phase, by considering MNIST (Deng, 2012), SVHN (Goodfellow et al., 2013), and CIFAR-10 (Krizhevsky et al., 2009), each comprising 10 distinct classes. Their use enables fair comparison with prior works while maintaining consistency with commonly adopted evaluation protocols in the literature.

Moreover, *CapsNets* are known to incur substantially higher computational complexity and routing overhead compared to conventional CNN architectures, which partly explains why many existing *CapsNet* studies are evaluated on similar small benchmark datasets such as MNIST, SVHN, or CIFAR. In particular, *CapsNets* face significant challenges in scaling to large-scale datasets such as ImageNet. This limitation derives from several factors: the iterative routing-by-agreement mechanism introduces substantial computational cost that grows with the number of capsules and classes; large-scale datasets exhibit high variability in object pose, appearance, and background, which can destabilize *part-whole* agreement; and the absence of widely optimized and efficient implementations further exacerbates scalability constraints. Therefore, these factors

---

[1]`https://github.com/STawalbeh/Visual-Explanations-for-Capsule-Networks`

make *CapsNets* more difficult to scale compared to more efficient architectures such as *CNNs* and Transformers.

We further clarify that *Caps-CAM* can generalize to larger-scale datasets and more diverse visual domains. Since *Caps-CAM* is designed as a post-hoc explainability framework that does not modify the underlying *CapsNet* architecture or routing procedure, its applicability is not inherently restricted to small-scale datasets, provided that a *CapNet* can be effectively trained on the target data.

**CapsNet Architectures.** We evaluate the proposed method on top of various *CapsNet* architectures, including Dynamic Routing (*DR*) (Sabour et al., 2017), which comprises convolutional, primary capsule, and class capsule layers. In addition, we also employ the Matrix Capsules (*EM*) architecture (Hinton et al., 2018), which incorporates additional layers, such as multiple convolutional capsule layers. We also modified the *DR* architecture by incorporating three additional convolutional layers before the primary capsule layer, resulting in the *DR-Conv* variant. Finally, we integrated a transformer block before the primary capsule layer (*DR-Tra*). These modifications are expected to enhance feature representation. These architectures were trained until they achieved competitive accuracy on the validation sets, demonstrating performance comparable to that reported in the literature (Sabour et al., 2017; Hinton et al., 2018; Xi et al., 2017; Xiong et al., 2019; Yang et al., 2020)(Table. 1). We evaluated the performance on the training and validation sets, with the results reported in the supplementary material (see Table A1). For reproducibility purposes, we provide two tables (Tables A3 and A4 in the supplementary material) detailing the used *CapsNet* architectures. This offers a comprehensive overview of the input and output shapes for both the *encoder* and *decoder* components. Additionally, we provide a summary of the number of parameters used for each architecture (see Table A3 in the supplementary material). This provides insight into the complexity and capacity of each *CapsNet* variant, which is an important consideration in discussions of explainability, as more complex networks often present greater challenges for classification tasks (Zou et al., 2024; Noor et al., 2024).

**Explanation Methods.** For the implementation of *Caps-CAM* we consider the top-2 most relevant capsules, denoted as $\mathbf{I}_{top}(\mathbf{A}^*_{R,top} = \mathbf{A}^k_R[\mathbf{I}_{top}])$ (Sec. 3.3). The reason for selecting only 2 capsule dimensions out of the 8 dimensions is that, in *CapsNets*, each capsule dimension encodes a distinct characteristic, whose relevance can be quantified via its activation strength. Focusing on the entire capsule dimensions can produce diffuse or overlapping activations, which may weaken the resulting explanation map. Considering only these specific capsule dimensions will help focus on the regions that are genuinely relevant when evaluating the generated explanation heatmaps. In line with this, we apply the top-2 capsules to dynamic routing-based *CapsNets*. We adopt the same methodology for the *EM* routing-based architecture with minor modifications due to differences in the respective routing mechanisms. Specifically, *EM* operates with $4 \times 4$ (16) dimensions rather than 8 dimensions; from these, we selected 2 dimensions for analysis. We conducted an ablation experiment to justify the choice of employing only a subset of 2 capsule dimensions. Please see the supplementary material for more details (Figures [A7-A14]).

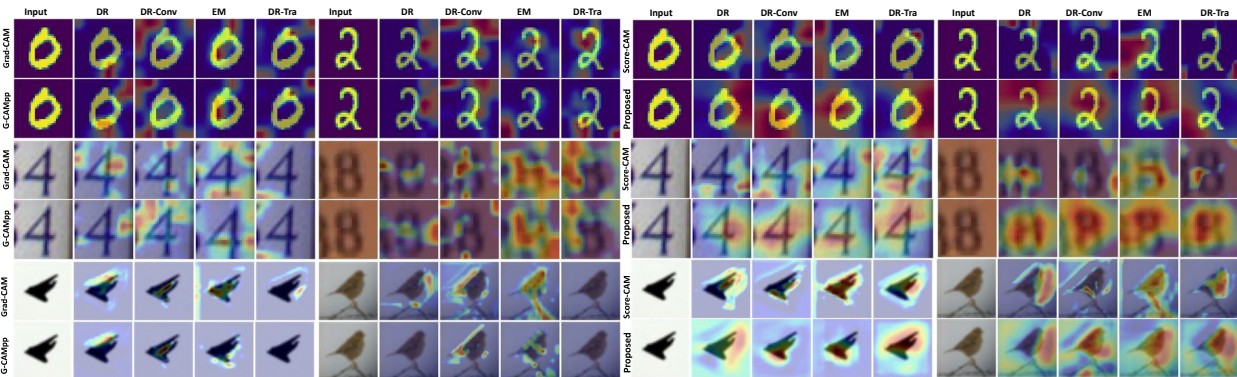

Figure 2: Heatmap visualizations generated by *Grad-CAM*, *Grad-CAM++*, *Score-CAM*, and *Caps-CAM*, highlighting the relevant regions within the MNIST (top), SVHN (middle), and CIFAR (bottom) datasets in the context of classification tasks(predicted correctly).

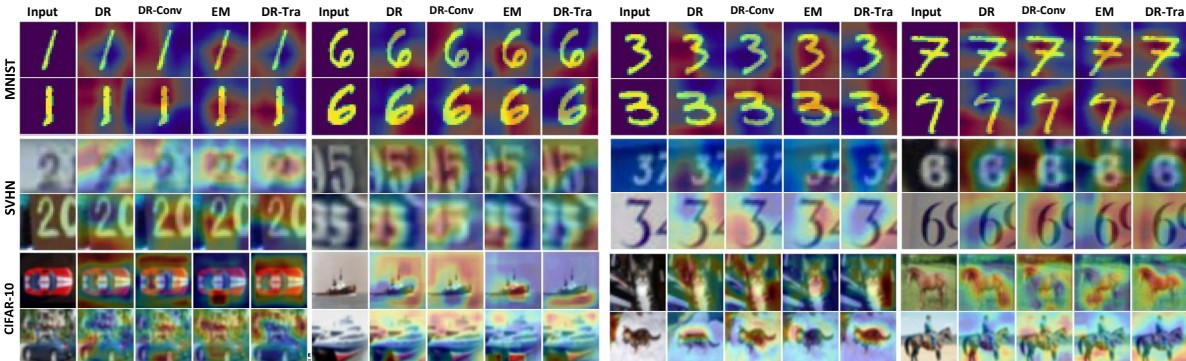

Figure 3: Additional *Caps-CAM* visual heatmaps for correct predictions across datasets (see Fig.A3 in the supplementary material for larger version).

We position our method w.r.t. several CAM-based explanation methods, i.e. *Grad-CAM* (Selvaraju et al., 2017), *Grad-CAM++* (Chattopadhay et al., 2018), and *Score-CAM* (Wang et al., 2020a), which we adapted to make them applicable to *CapsNets*. To achieve this, we reorganize the capsule dimension $Z$ as described in detail in Section 3.2. The matrix $\tilde{A}_l^k(x_i)$ is then used as input for the computation of both methods.

### 4.1 Qualitative Analysis.

This experiment aims to qualitatively compare the visual heatmaps produced by the proposed *Caps-CAM* method (Sec. 3.3) w.r.t. those from the competing explanation methods. We present some examples in Fig. 2 across the used datasets.

**Discussion:** As can be noted, the heatmaps produced by *Caps-CAM* exhibit fewer random artifacts and a smoother appearance compared to those generated by other methods, e.g., *Grad-CAM* and *Grad-CAM++*. Moreover, the visualizations produced by the proposed method seem to focus more on object-centric features. For example, in the MNIST dataset (Fig. 2/top), the heatmaps [0, 2] focus on the digit regions while suppressing noise in the background. Similarly, for SVHN (Fig. 2/middle), the heatmaps [4, 8] effectively highlight the digit itself (center parts) while ignoring irrelevant regions (background pixels). In CIFAR-10 examples [airplane, bird] (Fig. 2/bottom), *Caps-CAM* emphasizes key features such as the body of an airplane and the body of a bird, showing the focus of the network on object-centric features. In contrast, as noticed, the heatmaps generated by *Grad-CAM* and *Grad-CAM++* appear sparser compared to those produced by *Score-CAM* and *Caps-CAM*. This difference can be attributed to the incorporation of input masking steps, which may have two effects. On the one hand, since *Score-CAM* computes the weight of each feature map from the class score obtained after masking the input, denser activation maps preserve more information during the masking phase. This typically results in higher class scores in Eq. 3, which in turn favors these denser maps in the final heatmap generation (see Sec. 3.1). On the other hand, this process may capture spatially complementary information and by aggregating contributions from several relevant regions in the context of classification tasks, a denser heatmap is produced.

A second trend can be observed in Fig. 3, which shows additional examples produced by *Caps-CAM*. In these examples, *Caps-CAM* highlights different relevant features of the input images. There, we notice that for the MNIST/SVHN datasets, it focuses on the background [MNIST-6/SVHN-2,3,6], while for CIFAR-10, the method highlights the sky and the sea [boat] as well as the sky and the ground [horse/bottom class], where these features contribute to the classification output. These observations suggest that, in the context of classification tasks, the network considers background features as relevant cues for its decision-making phase.

Fig. 4 presents visual heatmaps generated by *Caps-CAM* for cases in which *CapsNets* produced incorrect predictions. For MNIST, the network often confuses visually similar classes, and the corresponding explanations show that attention is concentrated on regions shared across these classes. For example, as shown

in Fig. 4, the digits [4, 7, 9]/right exhibit overlapping shape patterns and the produced visual heatmaps highlight these shared components, while failing to emphasize discriminative regions that could resolve the ambiguity. For digit [3]/right, the heatmaps exhibit covered regions in the upper curved regions, which are also characteristic of digits [5] and [8]. This indicates that the model might rely on partial shape cues rather than globally discriminative features, resulting in misclassification. We also note a similar behavior when the model confuses classes with similar characteristics in digits [2, 3, 4, 7]/SVHN, where the heatmaps highlight non-object-centric features (i.e., background) of these digits. Furthermore, in both *DR-Conv* and *DR-Tra*, the network confuses digit [9]/SVHN, likely due to the presence of additional numbers within the input images, as it is covered by the heatmap. In some cases, the visual explanations highlight regions of the ground-truth digit despite incorrect predictions. For instance, in [3,4]/MNIST, the heatmaps emphasize the upper curved region of digit [3], which closely resembles the top loop structure of digit [8], the predicted class. At the same time, the highlighted central stroke of digit [4] corresponds to a stroke configuration commonly associated with digit [9], providing a visual explanation for the model's confusion. Similarly, in digits [3,7]/SVHN, the visual explanations focus on the central stroke of digit [3] and on the central and lower strokes of digit [7]. Notably, the elongated lower stroke of digit [7] resembles the vertical structure of digit [1], which helps explain why the model confuses these classes. For CIFAR-10, a higher degree of confusion between classes, which the visual explanation maps clearly reveal. For example, the visual heatmaps often highlight background regions [sky, grass, or street] that are shared across different categories, such as [airplane, car], [airplane, ship], [deer, horse], and [ship, frog]. Looking at the EM column in the MNIST section, several precise observations can be made. The visual heatmap concentrates on the vertical stroke of digit [1], which is its most salient feature. The prediction is digit [2] is understandable in this case since both share a rightward-leaning top stroke; however, the heatmap does not highlight the curved base that defines digit [2]. This explains that the network sees the correct digit but maps it to the incorrect class. Looking at class [3] and pred 8 case, the visual heatmap shows strong activation on both the upper and lower curved arcs of digit [3]. This reflects the shared stacked curved structures between the ground-truth and the prediction digit [8]. However, the heatmap does not emphasize the open right side that distinguishes digit [3] from digit [8]. This suggests that the model focuses on shared patterns rather than the key discriminative feature. For both digits 7 and [9] cases show visual heatmaps focused on vertical and diagonal strokes, where the misclassification in each case corresponds to the digit that most shares that dominant stroke direction. Similarly, in SVHN, when a ground-truth digit [2] is misclassified as [5], the heatmap emphasizes the upper curved region, which is a defining feature of digit [2]. For the case of CIFAR-10, the visual heatmaps show a pattern of diffuse, spatially scattered activations across all examined misclassification cases. For example, class (airplane-ship), the activation lacks any tight focal point, which suggests the network has no

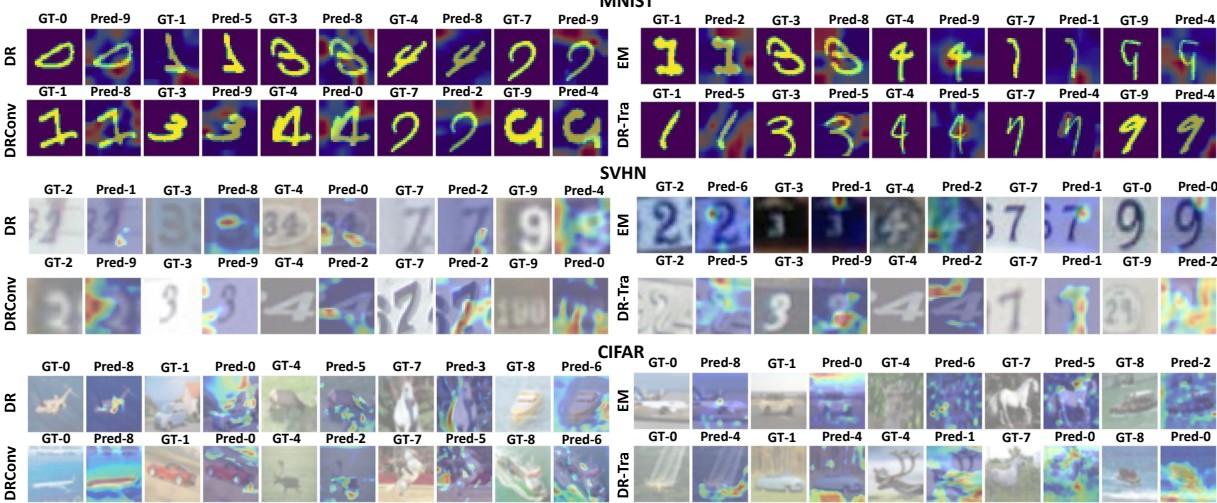

Figure 4: Additional heatmaps generated by the *Caps-CAM* method when *CapsNets* produced incorrect predictions across the evaluated datasets. For CIFAR, (0-airplane), (1-car),(4-deer), (7-horse),(8-ship).

well-defined internal representation of the airplane's distinguishing regions, with the confusion likely driven by shared contextual features such as elongated silhouettes and sky/water backgrounds.

The observations above suggest that *Caps-CAM* is capable of identifying both object-specific and context-specific features, which may be important for the overall prediction of the network. In addition, we observed differences in the visual explanations generated by *DR* (Sabour et al., 2017) and *EM* (Hinton et al., 2018) routing-based algorithms. We conjecture that these differences might be related to how the routing mechanisms define *part-whole* relationships, the characteristics of the routing algorithm, and the complexity of the dataset. These differences result in different characteristics of the explanation heatmaps. For instance, DR-based heatmaps tend to emphasize the most salient parts of an object that maximize agreement across capsules. In contrast, EM-based visual heatmaps appear smoother and more centered around the object or digit, as EM routing employs the clustering algorithm. Additional observation regarding EM architecture, the learned representations capture meaningful local features, even when the final prediction is incorrect. This behavior can be attributed to the clustering-based nature of EM routing, which aggregates information across capsules to form class representations. While this aggregation leads to smoother and more spatially coherent activations, it can also blur discriminative boundaries between classes, which makes the relevance estimation less selective. As a result, the final class assignment may not fully align with the most salient features highlighted in the heatmaps. This reveals a mismatch between feature-level representations and final decision-making under EM routing.

**Top-$Z$ Selection.** To further justify the design choice of selecting the top-$Z$ capsules, Fig. 5 presents qualitative comparisons of heatmaps generated using different numbers of top-ranked capsules (top-2, top-4, top-6, and top-8). A consistent trend can be observed across the examples; selecting a small number of top capsules (i.e., top-2) produces more focused and less noisy visual explanations (i.e., Top-2/MSTAR), whereas increasing the number of selected capsules leads to more diffuse heatmaps that include features related to other classes or less relevant to the predicted class. Beyond this observation, the behavior provides insight into how information is distributed within capsule representations. In particular, the results suggest that only a small subset of capsules contributes to the final decision, while the remaining capsules encode complementary or lower-confidence features. This observation is consistent with the inner-workings of *CapsNets*, where agreement mechanisms facilitate the selective activation of a small number of capsules for a given input. As more capsules are included, the heatmaps expand to cover additional regions (e.g., SVHN and CIFAR-10). This highlights a trade-off between compactness and completeness where selecting a small number of capsules yields focused explanations, while including more capsules captures additional context at the cost of introducing less relevant features. Furthermore, the consistency of the relevance ranking indicates that the selection of the top-2 capsules is stable and not arbitrary, but instead reflects the underlying importance of these capsules in the decision-making process. By restricting the visual heatmaps to the most relevant capsules, *Caps-CAM* filters out these weaker contributions, acting as an implicit denoising mechanism.

## 4.2 Quantitative Analysis

We quantify the validity of the generated explanation heatmaps to the decision-making process followed by the considered *CapsNets* models using the Average % Drop and the Area Over the Perturbation Curve (*AOPC*) metrics. A brief overview of these metrics is provided here; detailed definitions and implementation procedures are available in the supplementary material (Eq. A1 and Eq. A2). Following prior efforts (Chattopadhay et al., 2018; Samek et al., 2016), we considered only the correctly predicted inputs when computing both metrics. This is motivated by the fact that explanations for misclassified classes would reflect spurious evidence supporting the incorrect class, and are therefore less informative for evaluation purposes.

**Average % Drop (avg % Drop) (Chattopadhay et al., 2018).** This metric evaluates the reliability of an explanation by quantifying the change in the prediction confidence of the model after masking the input according to a class-specific visual heatmap. When the heatmap accurately identifies regions that are relevant to the decision of the model, suppressing less informative areas results in little or no reduction in confidence, yielding a low Average % Drop. A lower avg % drop indicates a better explanation heatmap.

**Average % Increase in Confidence (avg % Increase).** This metric evaluates the effectiveness of an explanation by measuring the change in the model's prediction confidence after enhancing the input

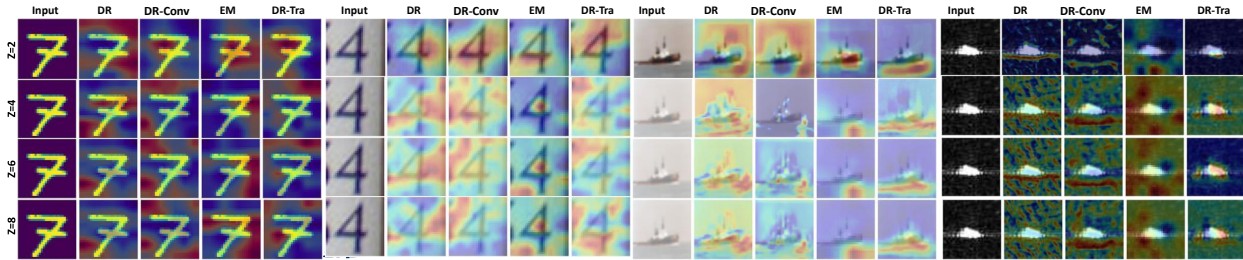

Figure 5: Visual examples for the proposed Caps-CAM explanation method across MNIST, SVHN, CIFAR-10, and MSTAR datasets. The visual heatmaps are shown for different values of $Z$, as $Z$ is the number of capsule dimensions.

based on a class-specific visual heatmap (e.g., by retaining the most relevant regions while suppressing less informative ones). When the heatmap highlights the regions important to the model's decision, focusing the input on these regions increases prediction confidence. A higher increase indicates that the explanation successfully captures discriminative evidence used by the model. Consequently, a higher avg % increase reflects a better-quality explanation heatmap.

**Area Over the Perturbation Curve ($AOPC$) (Samek et al., 2016).** This metric quantifies the average change in model output in response to progressive input perturbations. Following the work of Gu & Tresp (2021), we applied a similar perturbation setting to the generated explanation heatmaps using a progressive perturbation strategy guided by the explanation heatmaps produced by the method being evaluated. At each perturbation step, a $5 \times 5$ patch centered at the most relevant unperturbed spatial location is replaced with random noise. Once a location is perturbed, it is excluded from subsequent steps. This procedure is repeated for a fixed number of (15) perturbation steps. A higher $AOPC$, implies a better explanation heatmap, as it implies that the most relevant locations identified by the heatmap correspond to the information that is most relevant to the features exploited by the network.

For easier comparison between the different configurations, explanation heatmaps are produced for correctly predicted labels, and the methods are evaluated accordingly. Table 2 reports the mean percentage drop scores averaged across all classes for each dataset and evaluated architecture.

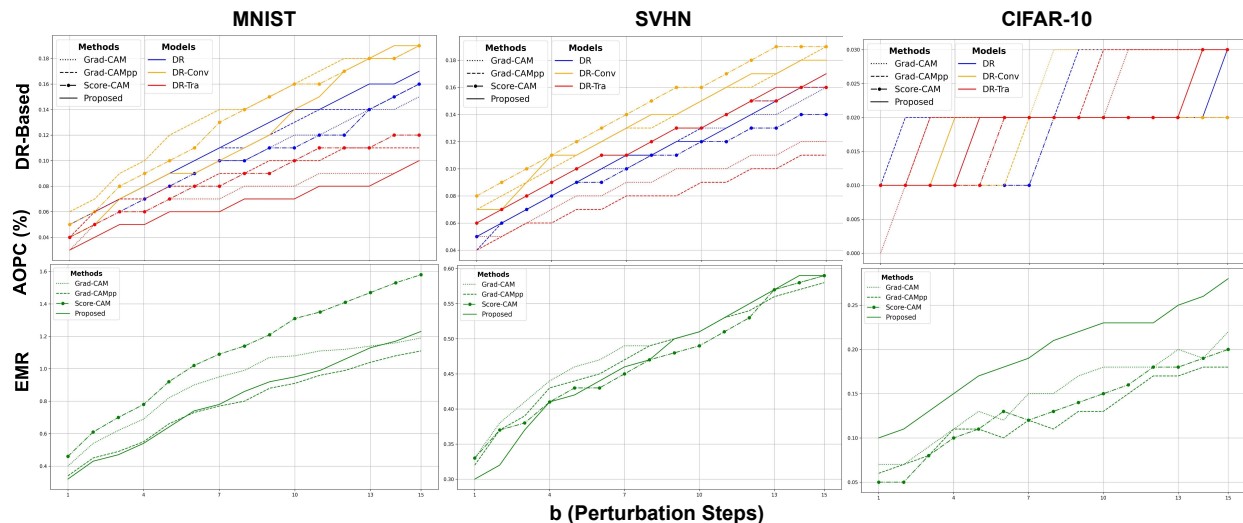

Figure 6: AOPC (%) values computed over 15 perturbation steps for visual explanations generated by various explanation methods applied to *CapsNet* architectures across the datasets. Refer to the appendix for the actual numbers (Tables A5, A6 and A9).

| Dataset | MNIST | | | | SVHN | | | | CIFAR-10 | | | | MSTAR | | | |
|---|---|---|---|---|---|---|---|---|---|---|---|---|---|---|---|---|
| Method | DR | DR-Conv | DR-Tra | EM | DR | DR-Conv | DR-Tra | EM | DR | DR-Conv | DR-Tra | EM | DR | DR-Conv | DR-Tra | EM |
| Grad-CAM | 30 | 33 | 23 | 31 | 46 | 51 | 44 | 31 | 45 | 47 | 46 | 43 | 52 | 53 | 52 | 42 |
| Grad-CAM++ | 27 | 27 | 21 | 32 | 48 | 49 | 43 | 31 | 43 | 43 | 43 | 47 | 46 | 51 | 51 | 38 |
| Score-CAM | 0.3 | 1 | 15 | **11** | 14 | 18 | 13 | 33 | 17 | 19 | **14** | 47 | 21 | 27 | 33 | **19** |
| Caps-CAM (Proposed) | **0.2** | **0.6** | **14** | 46 | **3** | **9** | **9** | **31** | **14** | **17** | 15 | **35** | **3** | **1** | **21** | 37 |

Table 2: Average % drop (lower is better) for different explanation methods across MNIST, SVHN, CIFAR-10, and MSTAR datasets. The models compared are Dynamic Routing (*DR*), DR with deep *conv* layers (*DR-Conv*), DR with transformer blocks before the *PC* layer (*DR-Tra*), and EM-routing (*EM*).

| Dataset | MNIST | | | | SVHN | | | | CIFAR-10 | | | | MSTAR | | | |
|---|---|---|---|---|---|---|---|---|---|---|---|---|---|---|---|---|
| Method | DR | DR-Conv | DR-Tra | EM | DR | DR-Conv | DR-Tra | EM | DR | DR-Conv | DR-Tra | EM | DR | DR-Conv | DR-Tra | EM |
| Grad-CAM | 12 | 16 | 4 | 11 | 1 | 2 | 2 | 11 | 6 | 2 | 1 | 20 | 4 | 1 | 11 | 12 |
| Grad-CAM++ | 17 | 2 | 4 | 23 | 1 | 2 | 2 | 22 | 6 | 3 | 3 | 17 | 5 | 1 | 5 | 10 |
| Score-CAM | 83 | 72 | 5 | 19 | 19 | 19 | 15 | 7 | 18 | 20 | 22 | 4 | 14 | 9 | 11 | 24 |
| Caps-CAM (Proposed) | 86 | 96 | 5 | 23 | 34 | 39 | 16 | 22 | 18 | 19 | 19 | 5 | 20 | 38 | 13 | 20 |

Table 3: Average increase in confidence (higher is better) for different explanation methods across MNIST, SVHN, CIFAR-10, and MSTAR datasets. The models compared are Dynamic Routing (*DR*), DR with deep *conv* layers (*DR-Conv*), DR with transformer blocks before the *PC* layer (*DR-Tra*), and EM-routing (*EM*).

**Discussion:** When focusing on ***avg % Drop***, notably, the *DR* model achieves the best performance when utilizing *Caps-CAM* across the datasets. Table 2 shows that *Caps-CAM* consistently outperforms other well-known methods across all architectures and datasets, with two exceptions. Specifically, *Caps-CAM* shows a higher average drop on both the *EM* architecture for the MNIST dataset and on the *DR-Tra* architecture for the CIFAR-10 dataset. While the heatmaps visually align with the digits in MNIST (Fig. 2/top and Fig. 3/top), the high average drop indicates that the highlighted regions alone may not fully justify the predictions made by the model. This suggests that the model may rely on additional patterns or background cues not captured by the heatmaps. These observations underscore the importance of explanation methods that are reliable both visually and quantitatively. Visually reliable explanations highlight the regions that the model genuinely relies on when making a prediction, whereas quantitatively reliable explanations are reflected in metrics such as Average % Drop, with lower values indicating higher explanation quality. Considering both criteria together provides a more comprehensive evaluation of the reliability of the generated heatmaps. Addressing this gap can significantly enhance the reliability of the proposed method.

Compared to gradient-based methods, *Caps-CAM* clearly outperforms *Grad-CAM* and *Grad-CAM++*. This improvement stems from two key factors. First, the masking of the input using relevance activation maps ensures that only the most informative regions of the input contribute to the explanation, which reduces noise from irrelevant regions. Second, the use of calculated confidence scores prioritizes capsules that are most predictive of the output, further emphasizing important regions. Together, these mechanisms enhance the reliability of the visual heatmaps, which explains the improved performance compared to *Grad-CAM* and *Grad-CAM++*.

**Effect of Top-$Z$ Selection (*Avg % drop*).,** table 4 summarizes the Avg % drop for different values of $Z$. The results show that the performance of *Caps-CAM* remains stable across different values of $Z$, with only minor variations observed for DR-based architectures. Notably, selecting a small number of capsules (e.g., $Z = 2$) achieves comparable/better performance than larger values, which supports the stability of the top-$Z$ selection strategy. In contrast, EM-based models exhibit a noticeable reduction as $Z$ increases, which suggests that incorporating additional capsules introduces less relevant/noisy contributions. These results confirm that restricting the explanation to a small set of relevant capsules leads to discriminative visual heatmaps, while also improving the robustness of the proposed method w.r.t the choice of $Z$.

When focusing on ***avg % Increase in Confidence***, notably, the *DR* and *DR-Conv* models achieve the best performance when utilizing *Caps-CAM* across the datasets (similar story as avg % drop). Similar to avg % drop and based on Table 3, *Caps-CAM* outperforms or achieves competitive results compared to other well-known methods across all architectures and datasets, with a few exceptions. Specifically, *Caps-CAM* yields a lower average confidence increase on the *DR-Tra* architecture for the MNIST dataset compared to Score-CAM, and on the *EM* architecture for the CIFAR-10 dataset, where it achieves the lowest increase

| Dataset | MNIST | | | | SVHN | | | | CIFAR-10 | | | | MSTAR | | | |
|---|---|---|---|---|---|---|---|---|---|---|---|---|---|---|---|---|
| Method | DR | DR-Conv | DR-Tra | EM | DR | DR-Conv | DR-Tra | EM | DR | DR-Conv | DR-Tra | EM | DR | DR-Conv | DR-Tra | EM |
| $Z = 2$ Caps-CAM (Proposed) | 0.2 | 0.6 | 14 | 46 | 3 | 9 | 9 | 31 | 14 | 17 | 15 | 35 | 3 | 1 | 21 | 37 |
| $Z = 4$ Caps-CAM (Proposed) | 0.1 | 1 | 12 | 83 | 3 | 2 | 9 | 34 | 13 | 17 | 15 | 27 | 3 | 1 | 18 | 50 |
| $Z = 6$ Caps-CAM (Proposed) | 0.2 | 1 | 13 | 84 | 3 | 3 | 9 | 34 | 14 | 17 | 15 | 26 | 3 | 1 | 19 | 50 |
| $Z = 8$ Caps-CAM (Proposed) | 0.2 | 1 | 12 | 86 | 3 | 2 | 9 | 35 | 14 | 14 | 15 | 26 | 3 | 1 | 20 | 47 |

Table 4: Average % drop (lower is better) for the proposed Caps-CAM explanation method across MNIST, SVHN, CIFAR-10, and MSTAR datasets. Models compared are Dynamic Routing (*DR*), DR with deep convolutional layers (*DR-Conv*), DR with transformer blocks before the *PC* layer (*DR-Tra*), and EM-routing (*EM*). Results are shown for different values of *Z*, as *Z* is the number of capsule dimensions.

among all methods. While the heatmaps visually highlight the relevant regions in the input (Fig. 2/top and Fig. 3/top), the joint evidence from both metrics confirms that *Caps-CAM* captures the features the model genuinely relies on when making its predictions. This consistency between the two complementary metrics suggests that *Caps-CAM* identifies the most discriminative input regions, which leads to both a minimal drop and a significant boost in model confidence when those regions are removed, respectively. These observations further underscore the importance of evaluating explanation methods through multiple quantitative perspectives. A low avg % drop indicates that the highlighted regions alone are sufficient to preserve the model's decision, while a high average % increase in confidence indicates that those same regions are the primary drivers of that decision. The notable performance of *Caps-CAM* on both metrics supports its behaviour as a post-hoc visual explanation method for *CapsNet* architectures and indicates that the generated heatmaps are well aligned with the model's predictions.

Regarding *AOPC*, similar to avg.% drop metric, the *AOPC* metric exhibits consistent behavior across most cases (Fig. 6), regardless of the explanation method or the *CapsNet* architecture used for evaluation (see Table A5/MNIST/DR, Table A6/SVHN/DR, Table A6/CIFAR-10/DR, and Table A9/bottom/EMR in the supplementary material). Across the datasets, *Caps-CAM* consistently outperforms other explanation methods (Table 5,6, and 9), with the exception of *Score-CAM*, which achieves higher *AOPC* scores when the *DR-Conv* (SVHN) (Fig. 6-top) (Table 6) and *EM* (MNIST) (Fig. 6-bottom) (Table 9) architectures are used to generate the visual explanations. Furthermore, *Caps-CAM* is specifically tailored for *CapsNets*, which facilitates producing visual explanations that leverage the relevant features, resulting in heatmaps that are more contextually aware compared to gradient-based methods originally designed for *CNN* architectures.

**Effect of Top-*Z* Selection (*AOPC*).** The *AOPC* results across different values of *Z* (top-2, top-4, top-6, and top-8) reveal a consistent trend. In most cases, selecting a small number of capsules (i.e., $Z = 2$) achieves the highest or competitive *AOPC* scores across datasets and architectures. This indicates that selecting the units identified by the top-2 capsule dimensions leads to stable performance in model confidence, which suggests that these capsules capture the relevant features driving the prediction. Furthermore, the results provide insight into the structure of capsule representations. In particular, they suggest that the most discriminative information is concentrated in a small subset of capsules, while additional capsules tend to encode complementary or less relevant features that might be related to other classes. More importantly, the performance remains stable across different values of *Z*, with only minor variations observed in most settings. Even in cases where $Z > 2$ does not achieve the best performance, the results remain competitive, indicating that the proposed method is not overly sensitive to the exact choice of *Z*. This robustness suggests that the relevance ranking provides a reliable ordering of capsule importance, and that the top-ranked capsules capture the core evidence used by the network.

### 4.3 Use Case: Moving and Stationary Target Acquisition and Recognition

To demonstrate the practical value of the proposed *Caps-CAM* method, we test it on a more realistic setting, specifically, Moving and Stationary Target Acquisition and Recognition from satellite images (Touafria &

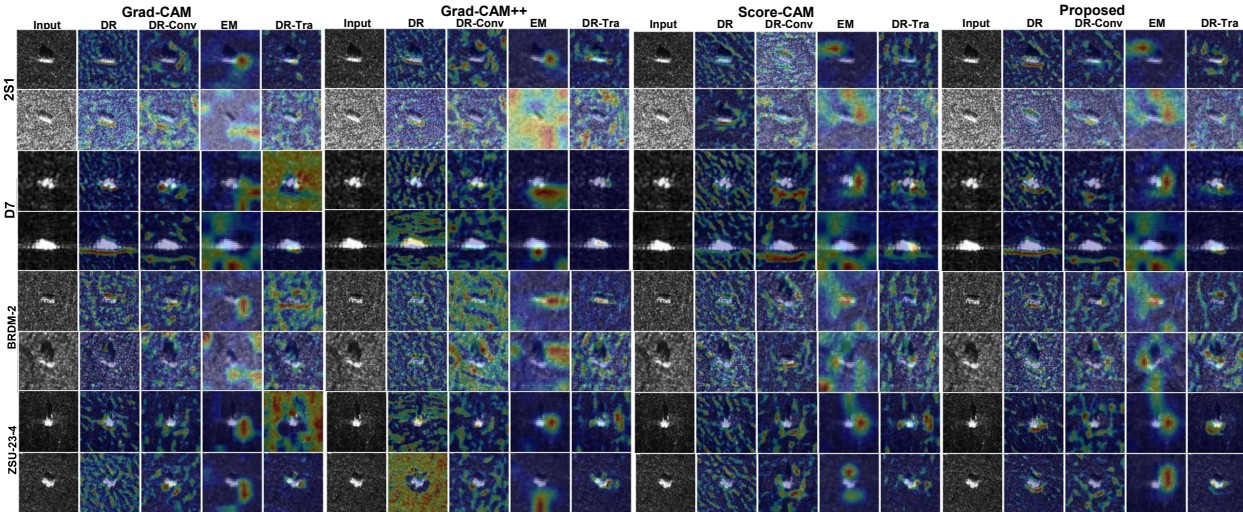

Figure 7: Heatmap visualizations generated by the proposed *Caps-CAM* method, compared to state-of-the-art methods, highlighting the relevant regions w.r.t classification problem within the MSTAR dataset (see Fig. A1 and A2 in the supplementary material for a larger version).

Yang, 2019; Dai et al., 2023). For this purpose, we conduct experiments on the MSTAR dataset.[2] Training on this dataset required data augmentation in order to mitigate overfitting. Please see the supplementary material for more details.

The following discussion presents qualitative examples of the generated explanations and quantitatively evaluates their validity considering the two previously introduced metrics, i.e., avg % drop and *AOPC*. For the latter, we report *AOPC* scores over 15 perturbation steps.

**Qualitative Analysis.** Fig. 7/right depicts examples of the explanation heatmaps generated by the proposed *Caps-CAM* method, alongside other methods from the literature, evaluated across several *CapsNet* architectures. As shown in Fig 7, the visual explanations produced by *Caps-CAM* generally exhibit fewer random artifacts compared to those generated by *Grad-CAM* and *Grad-CAM++*. This reflects that *Caps-CAM* leverages the structured capsule representations and relevance-weighted activations, which focus only on the capsules that contribute most to the predicted class. As a result, irrelevant regions are effectively suppressed, producing cleaner heatmaps that better reflect on the decision of the network. Furthermore, the proposed method provides an indication that the considered models focus not only on object-centric features, but on contextual features as well. This can be noticed in the case of class 2S1(top-right) (Fig. 7/right), where the visualizations focus on the shadow of the vehicle. A similar behavior is observed for the BRDM-2

---

[2]`https://www.sdms.afrl.af.mil/index.php?collection=`

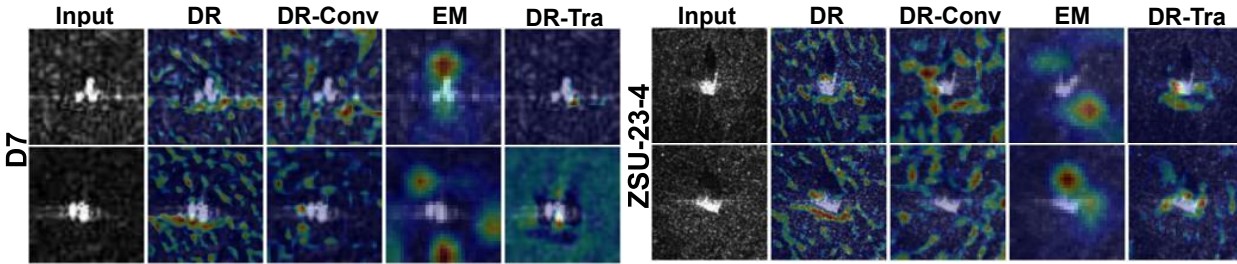

Figure 8: Additional explanation heatmaps generated by the proposed *Caps-CAM* method for heavy tank classes in the MSTAR dataset.

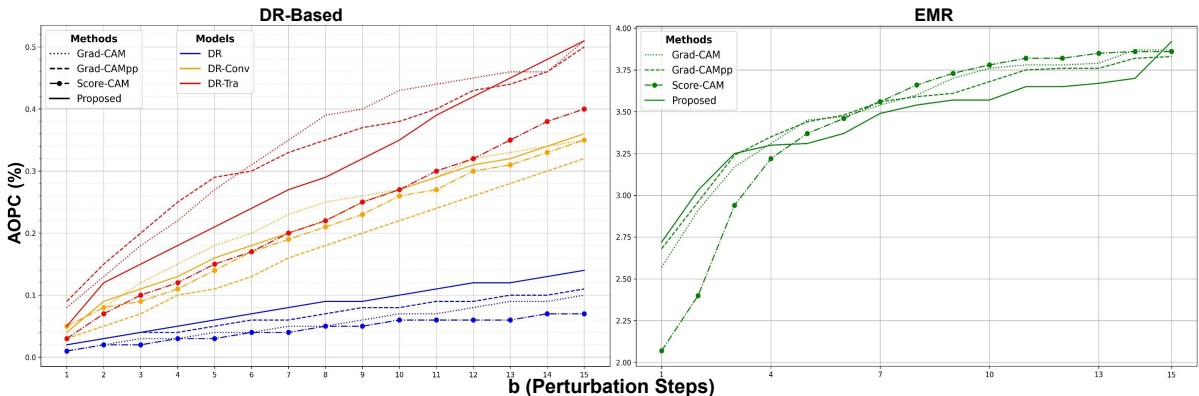

Figure 9: AOPC (%) values over 15 perturbation steps metric for visual explanation generated by explanation methods on *CapsNets* trained on the MSTAR dataset. Refer to the appendix for the actual numbers (Tables A8 and A9).

class using the *DR-Tra* architecture. Another notable observation is found in the D7 class (Fig. 7/right), where the visual explanations tend to focus on the ground pressure marks left by the vehicle, likely due to its significant weight. A similar behavior is observed in the ZSU-23-4 class. As both classes correspond to heavy military vehicles, this pattern supports our interpretation. Additional examples illustrating this behavior are provided in Fig. 8. More figures supporting our findings can be found in the supplementary material.

**Quantitative Analysis.** When focusing on the *avg % drop* metric, the proposed *Caps-CAM* method continues to demonstrate promising results across most architectures. As shown in Table 2, *Caps-CAM* achieves the lowest drop scores on the *DR*, *DR-Conv*, and *DR-Tra* architectures, outperforming all other baseline methods, including *Score-CAM*, *Grad-CAM*, and *Grad-CAM++*. However, an exception is observed with the *EM* architecture. where *Score-CAM* outperforms *Caps-CAM*. This aligns with the trend previously observed on MNIST, where *Caps-CAM*'s performance on the *EM* architecture was also less effective. This may result from the unique routing behavior in the *EM* variant, which impacts the reliability of *Caps-CAM*'s relevance scores. These results highlight the stability of *Caps-CAM* in dynamic routing-based capsule architectures, particularly when applied to complex real-world datasets such as MSTAR.

As shown in Table 3, *Caps-CAM* achieves the highest confidence increase on the *DR* and *DR-Conv* architectures, complementing the low drop scores observed for the same architectures in Table 2. Together, these two metrics consistently identify *DR* and *DR-Conv* as the architectures where *Caps-CAM* produces visual explanations that are aligned with the model predictions. The weaker performance on the *DR-Tra* and *EM* architectures is equally consistent across both metrics, further suggesting that the transformer blocks and *EM* routing variant introduce feature representations that are less suited to *Caps-CAM*'s relevance scoring. This coherence between the two complementary metrics reinforces the reliability of *Caps-CAM* as a good explanation method for dynamic routing-based capsule architectures on complex real-world datasets such as MSTAR.

When looking at the *AOPC* (Fig. 9), we observe that the *AOPC* curves for all explanation methods are relatively close for each architecture (see Table A8/DR and Table A9/top/EMR in the supplementary material for the actual scores). Nevertheless, *Caps-CAM* outperforms the other methods, despite not relying on the classification layer. This behavior may indicate that *Caps-CAM* explanations are generated using the most relevant capsules, identified through the product between activation maps and their corresponding gradients. This process highlights more relevant features that contribute to improved explanation quality.

## 5 Limitations of Visual Explanation Evaluation

It is important to note that evaluating visual explanation methods remains inherently challenging. In the absence of ground-truth feature importance, saliency-based evaluation cannot directly verify the correctness

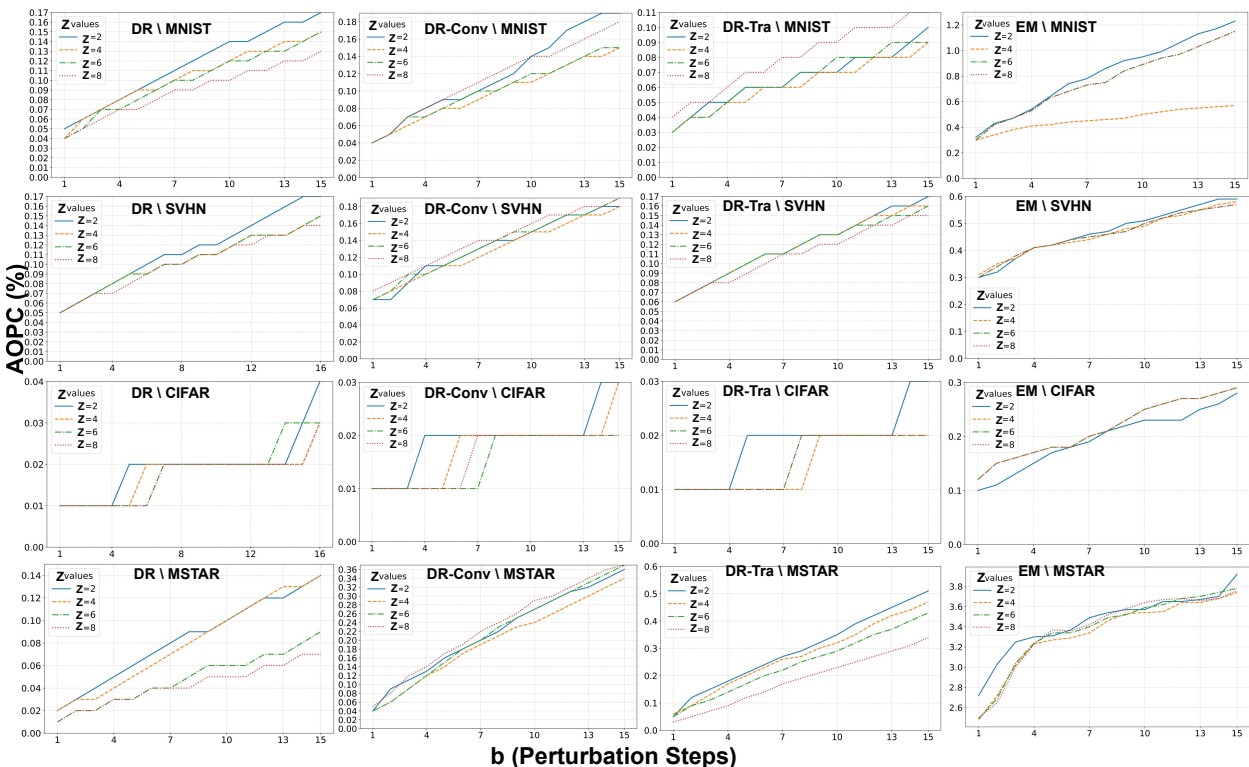

Figure 10: *AOPC* (higher is better) for the proposed Caps-CAM explanation method across MNIST, SVHN, CIFAR-10, and MSTAR datasets. Models compared are Dynamic Routing (*DR*), DR with deep convolutional layers (*DR-Conv*), DR with transformer blocks before the *PC* layer (*DR-Tra*), and EM-routing (*EM*). Results are shown for different values of $Z$, as $Z$ is the number of capsule dimensions.

of explanations. While perturbation-based metrics such as confidence drop and increase are widely used, they may be sensitive to the choice of perturbation strategy and potential distribution shift induced by masking/occlusion. Furthermore, visually plausible heatmaps do not necessarily guarantee that the highlighted regions fully correspond to the model's true decision process. Therefore, the quantitative and qualitative evaluations presented in this work should be interpreted as indicative measures of explanation quality rather than absolute validation of faithfulness Gevaert et al. (2024).

Another limitation of the conducted evaluation is that it does not include a direct comparison with explanation methods specifically designed to leverage routing information in capsule networks. While several routing-based methods produce explanations in a different representation space, which complicates a one-to-one quantitative comparison with visual saliency methods such as *Caps-CAM*, this difference should not be interpreted as evidence of equivalence. Consequently, the results presented in this work demonstrate the performance of *Caps-CAM* relative to the considered baselines but do not establish how it compares to routing-based explanation methods. A comprehensive evaluation across both visual attribution and routing-based explanation paradigms remains an important direction for future work.

## 6    Conclusion

We propose an explanation method, *Caps-CAM*, specifically designed for justifying the outputs of *CapsNet* architectures. Comparative analysis with existing state-of-the-art methods shows the effectiveness of *Caps-CAM*. More precisely, both qualitative visualizations and quantitative evaluations confirm that *Caps-CAM* outperforms other methods, in most cases by highlighting the most relevant regions of input images at different levels of importance in the context of classification tasks. A limitation of this work is that *Caps-*

*CAM* produces visual explanations specifically from capsule routing backbones. However, extending this method to other backbone architectures (e.g., graph-based backbones) remains unexplored and represents a potential direction for future research. *Caps-CAM* could be extended to multimodal tasks and focus on the methods for combining capsule relevance across modalities. Furthermore, the trade-off between explanation methods and model complexity requires further investigation to gain a more comprehensive understanding of the observed model behavior.

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
