# Visual Explanations for Capsule Networks Supplementary Material

**Saja Tawalbeh**
*University of Antwerp, sqIRL/IDLab, imec*

*saja.tawalbeh@uantwerpen.be*

**José Oramas**
*University of Antwerp, sqIRL/IDLab, imec*

*jose.oramas@uantwerpen.be*

**Reviewed on OpenReview:** *https://openreview.net/forum?id=eNQK9WJkid*

## 1 Introduction

This supplementary material presents additional details and experiments that further support and validate the method proposed (*Caps-CAM*). These supplementary results provide additional empirical evidence of the robustness of the proposed method and complement the findings discussed in this chapter.

## 2 CapsNet Architectures

We evaluate the proposed method on various *CapsNet* architectures, including Dynamic Routing (*DR*) Sabour et al. (2017), which comprises convolutional, primary capsule, and class capsule layers. Additionally, we modified the *DR* architecture by incorporating three additional convolutional layers before the primary capsule layer, resulting in the *DR-Conv* variant. Finally, we integrated a transformer block before the primary capsule layer (*DR-Tra*). This modification is expected to enhance feature representation. These architectures were trained until achieving a competitive accuracy (Table. A1). In addition to *DR*-based architectures, we also employ the Matrix Capsules (*EM*) architecture Hinton et al. (2018), which incorporates additional layers, such as multiple convolutional capsule layers.

## 3 Classification Performance

Several CapsNets were used for model training Table A1 summarizes the classification performance reported on the training, validation, and testing sets across the considered datasets and the used architectures.

Table A2 reports the approximate number of trainable parameters for each *CapsNet* variant across the four datasets used in this study: MNIST, SVHN, CIFAR-10, and MSTAR. As expected, models incorporating convolutional and transformer components (DR-Conv and DR-Tra) exhibit a significantly higher parameter

| Capsule Network | MNIST | | | SVHN | | | CIFAR-10 | | | MSTAR | | |
|---|---|---|---|---|---|---|---|---|---|---|---|---|
| | Train | Valid | Test | Train | Valid | Test | Train | Valid | Test | Train | Valid | Test |
| *DR* | 99.8 | 99.9 | 99.1 | 97.0 | 91.0 | 91.0 | 93.5 | 93.6 | 82.3 | 99.9 | 99.5 | 99.4 |
| *DR-Conv* | 99.8 | 99.8 | 99.0 | 99.6 | 92.4 | 90.5 | 89.0 | 89.0 | 74.0 | 100.0 | 99.4 | 99.7 |
| *DR-Tra* | 99.9 | 99.4 | 99.4 | 96.0 | 92.0 | 91.0 | 85.9 | 85.8 | 76.0 | 99.8 | 99.1 | 99.3 |
| *EM* | 99.1 | 98.1 | 98.0 | 82.7 | 81.4 | 80.0 | 73.8 | 73.4 | 72.2 | 83 | 87.9 | 86.6 |

Table A1: *CapsNets* performance in terms of accuracy across the MNIST, SVHN, CIFAR-10, and MSTAR military target datasets. Dynamic Routing (DR), Dynamic Routing with deep convolutional layers (DR-Conv), Dynamic Routing with transformer blocks added before the *PC* layer (DR-Tra), and EM-routing (EM).

| CapsNets | MNIST | SVHN | CIFAR-10 | MSTAR |
|----------|-------|------|----------|-------|
| DR | 8.2M | 11.7M | 42.2M | 42.3M |
| DR-Conv | 9.4M | 12.9M | 208.5M | 208.6M |
| DR-Tra | 8.9M | 12.4M | 207.7M | 207.8M |
| EM | 6.8M | 7.9M | 89.1M | 88.9M |

Table A2: Approximate number of parameters in each *CapsNet* variant for different datasets. CIFAR-10 and MSTAR share the same input size and architecture; MSTAR differs only in the number of output classes ($Y = 8$), which slightly affects the EM decoder.

count, particularly for CIFAR-10 and MSTAR, due to the increased model complexity required to handle more diverse visual features. Notably, CIFAR-10 and MSTAR share the same input resolution and architectural design; however, the MSTAR variant includes fewer output classes, leading to a marginal reduction in the number of parameters for the EM-based model. Table A3 presents details of the CapsNet proposed by Sabour et al. (2017). This table shows more details of the input and output shapes in both *encoder* and *decoder* parts. Table A4 presents more details of Matrix capsules with the *EM* Hinton et al. (2018) variant. We show extra details of the input and output shapes in both the encoder and decoder parts.

# 4 Dataset

Following the literature Sabour et al. (2017); Hinton et al. (2018); Gu et al. (2021); Byerly et al. (2021); Tawalbeh & Oramas (2024); Mitterreiter et al. (2023), we adopt well-established datasets for the evaluation of *CapsNets*. More specifically, we consider the MNIST, SVHN, and CIFAR-10, each comprising 10 distinct classes. MNIST is a grayscale dataset depicting hand-drawn digits. It consists of 60k training images, from which 10k have been selected for validation, and 10k for testing. SVHN is an RGB dataset depicting number plates. It consists of 63k images for training, 10k for validation, and 26k for testing. The CIFAR-10 dataset is composed of 60k images. The dataset has been split into 40k images for training and 10k images for both the validation and the test phases.

The Moving and Stationary Target Acquisition and Recognition (MSTAR) dataset is a widely used standard in synthetic aperture radar (SAR) image processing, mainly designed for target detection and recognition tasks. It is compiled and processed by the Sandia National Lab and is publicly available [1]. The dataset consists of 8 types of military targets, comprising a total of 9,466 grayscale SAR images. The images are split into 5,679 for training, 1,893 for validation, and 1,894 for testing. The target classes, such as; self-propelled howitzer (2S1), armored reconnaissance vehicle (BRDM-2), Eight-wheeled personnel carrier (BTR-60), Tracked bulldozer (D7), Non-vehicle calibration (SLICY), tank (T62), military truck (ZIL-131), and Self-propelled anti-aircraft gun (ZSU-23-4), represent a mix of military vehicles and calibration objects. Each sample is a grayscale SAR image chip of size 128×128 pixels, collected using X-band radar at depression angles of 15° and 17°. The dataset provides full 360° aspect coverage, introducing substantial variation in target orientation and enabling robust evaluation of pose-invariant automatic target recognition models.

**Data Augmentation.** The data augmentation process used in this work was applied exclusively to the MSTAR dataset to help mitigate overfitting. Each image was resized to 64×64 pixels, except for those used with the *DR* network, and then subjected to random horizontal flipping and random rotation of up to 10 degrees to simulate variations in viewpoint. In addition, color jittering was applied to adjust brightness, contrast, saturation, and hue. Although the SAR images are grayscale, such transformations can simulate sensor noise or environmental variability. This improves the robustness to illumination and acquisition inconsistencies. Finally, all images were converted to tensors and normalized using a mean and standard deviation of 0.5.

---

[1] https://www.sdms.afrl.af.mil/index.php?collection=

## 5 Evaluation

This section presents visual explanation heatmaps generated by the proposed *Caps-CAM* method. Then, we quantify the validity of the generated explanation heatmaps to the decision-making process, followed by the considered *CapsNets* models using the Average % Drop and the Area Over the Perturbation Curve (*AOPC*) metrics.

### 5.1 Qualitative Analysis

We present representative examples of the generated heatmaps in Figs.A4 and A5 for the SVHN, CIFAR-10 datasets, respectively. Additionally, Fig. A6 illustrates further heatmaps produced by the proposed *Caps-CAM* method on the MSTAR dataset.

In addition, we justify the selection of only 2 capsules $\mathbf{I}_{top}(\mathbf{A}^*_{R,top} = \mathbf{A}^k_R[\mathbf{I}_{top}])$ out of the 8 available dimensions by noting that *CapsNets* facilitate the identification and retention of units that are truly relevant. Figs. A7, A8, A9, and A10 demonstrate that the relevant units effectively highlight the regions utilized by *CapsNets* during prediction.

### 5.2 Quantitative Analysis

We quantify the validity of the generated explanation heatmaps to the decision-making process followed by the considered *CapsNets* models using the Average % Drop and the Area Over the Perturbation Curve *AOPC* metrics. Following prior work (Chattopadhay et al., 2018; Samek et al., 2016), we considered only the correctly predicted inputs when computing both metrics. This is motivated by the fact that explanations for misclassified classes would reflect spurious evidence supporting the incorrect class, and are therefore less informative for evaluation purposes.

**Average % Drop (avg % Drop) (Chattopadhay et al., 2018).** This metric is computed by performing an element-wise multiplication between the class-specific explanation heatmap $H_c$ and the input image. This

| Layer | MNIST | | SVHN | | CIFAR-10 | |
|---|---|---|---|---|---|---|
| | Input Shape | Output Shape | Input Shape | Output Shape | Input Shape | Output Shape |
| ***Encoder / DR*** | | | | | | |
| Conv $\Rightarrow$ ReLU | [1, 28, 28] | [20, 20, 256] | [3, 32, 32] | [24, 24, 256] | [3, 64, 64] | [56, 56, 256] |
| ***Encoder / DR-Conv*** | | | | | | |
| Conv $\Rightarrow$ Conv | [1, 28, 28] | [20, 20, 256] | [3, 32, 32] | [24, 24, 256] | [3, 64, 64] | [56, 56, 256] |
| Conv $\Rightarrow$ Conv | [20, 20, 256] | [20, 20, 256] | [24, 24, 256] | [24, 24, 256] | [56, 56, 256] | [56, 56, 256] |
| Conv $\Rightarrow$ Conv | [20, 20, 256] | [20, 20, 256] | [24, 24, 256] | [24, 24, 256] | [56, 56, 256] | [56, 56, 256] |
| ***Encoder / DR-Tra*** | | | | | | |
| Conv $\Rightarrow$ ReLU | [1, 28, 28] | [20, 20, 256] | [3, 32, 32] | [24, 24, 256] | [3, 64, 64] | [56, 56, 128] |
| Conv $\Rightarrow$ TransformerBlock | [20, 20, 256] | [20, 20, 256] | [24, 24, 256] | [24, 24, 256] | [56, 56, 128] | [56, 56, 128] |
| ***Capsule Layers*** | | | | | | |
| PC $\Rightarrow$ CC | [6, 6, 32] | [1152, 8] | [8, 8, 32] | [2048, 8] | [24, 24, 32] | [147456, 8] |
| CC $\Rightarrow$ Decoder | [1152, 8] | [$Y$, 16] | [2048, 8] | [$Y$, 16] | [147456, 8] | [$Y$, 16] |
| ***Decoder*** | | | | | | |
| Linear1 $\Rightarrow$ ReLU | [1, 160] | [1, 512] | [1, 160] | [1, 512] | [1, 160] | [1, 512] |
| Linear2 $\Rightarrow$ ReLU | [1, 512] | [1, 1024] | [1, 512] | [1, 1024] | [1, 512] | [1, 1024] |
| Linear3 $\Rightarrow$ Sigmoid | [1, 1024] | [1, (1, 28, 28)] | [1, 1024] | [1, (3, 32, 32)] | [1, 1024] | [1, (3, 64, 64)] |

Table A3: Input/output shapes for different *CapsNet* architectures based on Dynamic Routing (DR). The *DR-Conv* variant adds three convolutional layers before the primary capsule layer (*PC*), while the *DR-Tra* variant includes a transformer block before *PC*. Here, $Y$ is the number of target classes, and *CC* denotes the class capsule layer.

| Layers | MNIST | | SVHN | | CIFAR-10 | |
|---|---|---|---|---|---|---|
| | Input Shape | Output Shape | Input Shape | Output Shape | Input Shape | Output Shape |
| *Encoder:* | | | | | | |
| Conv $\Rightarrow$ BatchNorm | [1, 28, 28] | [14, 14, 32] | [3, 32, 32] | [16, 16, 32] | [3, 64, 64] | [32, 32, 32] |
| BatchNorm $\Rightarrow$ ReLU | [14, 14, 32] | [14, 14, 32] | [16, 16, 32] | [16, 16, 32] | [32, 32, 32] | [32, 32, 32] |
| PC $\Rightarrow$ ConvC1 | [14, 14, 544] | [14, 14, 544] | [16, 16, 32] | [16, 16, 544] | [32, 32, 32] | [32, 32, 544] |
| ConvC1 $\Rightarrow$ ConvC2 | [14, 14, 544] | [6, 6, 544] | [16, 16, 544] | [7, 7, 544] | [32, 32, 544] | [15, 15, 544] |
| ConvC2 $\Rightarrow$ ClassC | [6, 6, 544] | [4, 4, 544] | [7, 7, 544] | [5, 5, 544] | [15, 15, 544] | [12, 12, 544] |
| ClassC $\Rightarrow$ Decoder | [4, 4, 544] | [512, $Y$, 17] | [5, 5, 544] | [$Y$, 17] | [12, 12, 544] | [$Y$, 17] |
| *Decoder:* | | | | | | |
| Linear1 $\Rightarrow$ ReLU | [1, 160] | [1, 512] | [1, 160] | [1, 512] | [1, 160] | [1, 512] |
| Linear2 $\Rightarrow$ ReLU | [1, 512] | [1, 1024] | [1, 512] | [1, 1024] | [1, 512] | [1, 1024] |
| Linear3 $\Rightarrow$ Sigmoid | [1, 1024] | [1, (1, 28, 28)] | [1, 1024] | [1, (3, 32, 32)] | [1, 1024] | [1, (3, 64, 64)] |

Table A4: The input and the output shapes of the considered capsule network architecture (EM Routing Hinton et al. (2018). $Y$ represents the number of the target classes in the datasets.

produces a masked image in which pixel intensities are weighted according to their relevance, as indicated by the explanation heatmap, which preserves regions deemed important for the prediction. Conversely, pixels with low relevance scores are attenuated, reducing the influence of less important regions on the output of the model. The resulting masked image is then passed through a forward propagation of the pretrained model $F$, yielding a modified prediction score. A valid explanation heatmap should assign higher values to regions that are critical for the decision made by the network. If the heatmap correctly highlights these important regions, masking the less relevant regions should result in little to no reduction of confidence of the model, whereas suppressing important pixels would lead to a noticeable decrease in the prediction score.

Overall, a lower average % drop indicates a better explanation heatmap. This is calculated using the following equation (Eq. A1).

$$avg.Drop = \frac{max(0, F(x_i)_c - F(H_c \odot x_i)_c}{F(x_i)_c} \tag{A1}$$

In this context, $F(x_i)_c$ represents the predicted score for class $c$ for a given input image $x_i$, while $F(H_c \odot x_i)_c$ denotes the predicted score for class $c$ obtained when the input image is masked with the explanation heatmap $H_c$. An increase in the prediction/confidence score indicates that conflicting information present in the input image has been removed in the modified image. This suggests that the method fails to highlight the conflicting information.

**Area Over the Perturbation Curve (AOPC) (Samek et al., 2016).** This metric quantifies the average change in model output in response to input perturbations. Specifically, it computes the average difference between the prediction on the original input and predictions on progressively perturbed versions, where pixels are successively removed based on their relevance. $AOPC$ is computed using Eq. A2, where a higher $AOPC$ score indicates a more effective explanation.

$$AOPC = \frac{1}{L+1} \sum_{b=1}^{L} \left( F\left(x^{(0)}\right) - F\left(x^{(b)}\right) \right)_{p^{(x)}} \tag{A2}$$

Here, $L$ denotes the number of perturbation steps, $F$ represents the pre-trained model, and $x^{(b)}$ refers to the input image after $b$ perturbation steps. Following Gu & Tresp (2021), we applied a similar perturbation setting to the generated explanation heatmaps using a progressive perturbation strategy guided by *Caps-CAM* relevance maps. At each perturbation step, a $5 \times 5$ patch centered at the most relevant unperturbed spatial location is replaced with random noise. Once a location is perturbed, it is excluded from subsequent steps. This procedure is repeated for a fixed number of perturbation steps ($L = 15$).

Table A10 presents the Average % drop (lower is better) for the proposed Caps-CAM explanation method across MNIST, SVHN, CIFAR-10, and MSTAR datasets. Models compared are Dynamic Routing (*DR*),

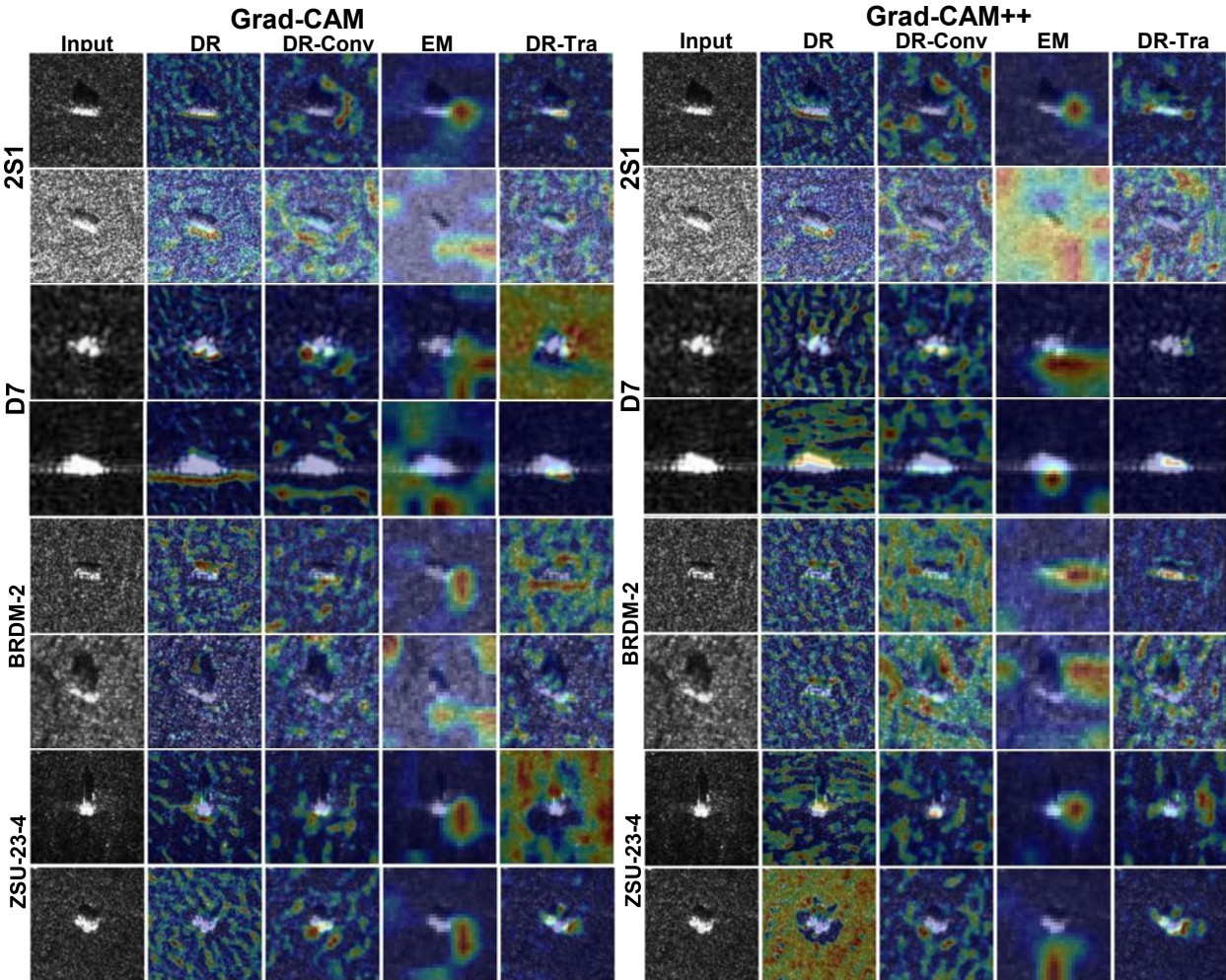

Figure A1: Heatmap visualizations generated by the proposed *Caps-CAM* method, compared to state-of-the-art methods, highlighting the relevant regions w.r.t classification problem within the MSTAR dataset.

DR with deep convolutional layers (*DR-Conv*), DR with transformer blocks before the *PC* layer (*DR-Tra*), and EM-routing (*EM*). Results are shown for different values of $Z$. Table A10 shows that *Caps-CAM* most of the cases outperforms other $Z$ values across all architectures and datasets

Regarding *AOPC*, Similar to the *avg.% Drop* metric, the *AOPC* metric demonstrates consistent behavior across most cases (Figures A11, A12, A13, and **??**), regardless of the chosen $Z$ value or the *CapsNet* architecture used for evaluation. This observation supports our choice of using only two capsule dimensions ($Z = 2$) as sufficient for generating meaningful explanation heatmaps. Selecting two dimensions enables the model to concentrate on the most informative features while avoiding redundant or diffuse activations that typically occur in higher-dimensional capsule spaces.

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

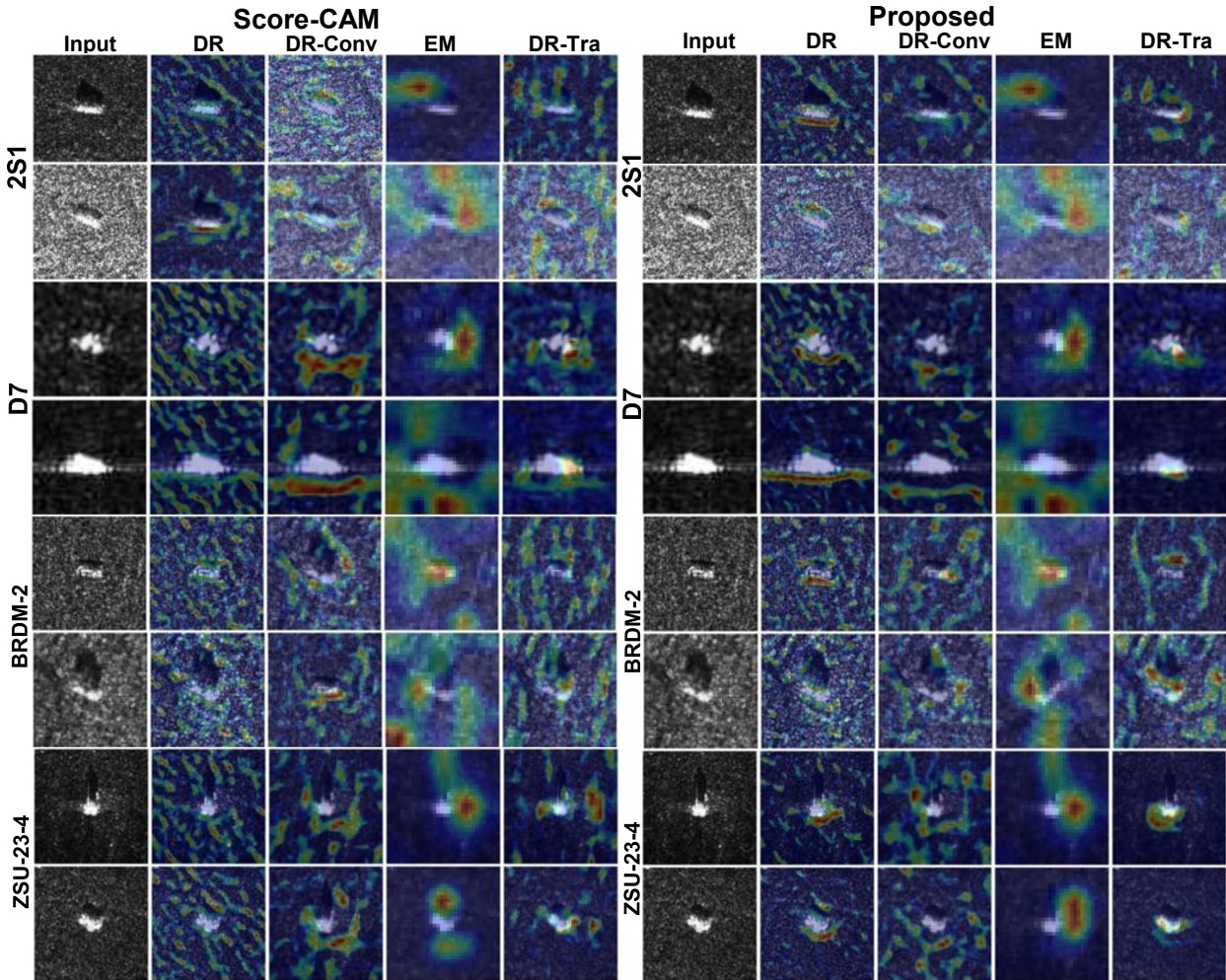

Figure A2: Heatmap visualizations generated by the proposed *Caps-CAM* method, compared to state-of-the-art methods, highlighting the relevant regions w.r.t classification problem within the MSTAR dataset.

Jindong Gu, Volker Tresp, and Han Hu. Capsule network is not more robust than convolutional network. In *CVPR*, 2021.

Geoffrey E Hinton, Sara Sabour, and Nicholas Frosst. Matrix capsules with em routing. In *ICLR*, 2018.

Matthias Mitterreiter, Marcel Koch, Joachim Giesen, and Sören Laue. Why capsule neural networks do not scale: Challenging the dynamic parse-tree assumption. In *AAAI*, 2023.

Sara Sabour, Nicholas Frosst, and Geoffrey E Hinton. Dynamic routing between capsules. *NeurIPS*, 2017.

Wojciech Samek, Alexander Binder, Grégoire Montavon, Sebastian Lapuschkin, and Klaus-Robert Müller. Evaluating the visualization of what a deep neural network has learned. *IEEE transactions on neural networks and learning systems*, 28(11):2660–2673, 2016.

Saja Tawalbeh and José Oramas. Towards the characterization of representations learned via capsule-based network architectures. *Neurocomputing*, 2024.

| CapsNet | Method | Perturbation Step $b$ | | | | | | | | | | | | | | |
|---|---|---|---|---|---|---|---|---|---|---|---|---|---|---|---|---|
| | | 1 | 2 | 3 | 4 | 5 | 6 | 7 | 8 | 9 | 10 | 11 | 12 | 13 | 14 | 15 |
| DR | Grad-CAM | 0.05 | 0.06 | 0.07 | 0.08 | 0.09 | 0.09 | 0.10 | 0.10 | 0.11 | 0.12 | 0.12 | 0.13 | 0.14 | 0.14 | 0.15 |
| | Grad-CAM++ | 0.05 | 0.06 | 0.07 | 0.08 | 0.09 | 0.10 | 0.11 | 0.11 | 0.12 | 0.13 | 0.14 | 0.14 | 0.14 | 0.15 | 0.16 |
| | Score-CAM | 0.04 | 0.05 | 0.06 | 0.07 | 0.08 | 0.09 | 0.10 | 0.10 | 0.11 | 0.11 | 0.12 | 0.12 | 0.14 | 0.15 | 0.16 |
| | Proposed | 0.05 | 0.06 | 0.07 | 0.08 | 0.09 | 0.10 | 0.11 | 0.12 | 0.13 | 0.14 | 0.14 | 0.15 | 0.16 | 0.16 | 0.17 |
| DR-Conv | Grad-CAM | 0.06 | 0.07 | 0.09 | 0.10 | 0.12 | 0.13 | 0.14 | 0.14 | 0.15 | 0.16 | 0.17 | 0.18 | 0.18 | 0.19 | 0.19 |
| | Grad-CAM++ | 0.06 | 0.07 | 0.09 | 0.10 | 0.12 | 0.13 | 0.14 | 0.14 | 0.15 | 0.16 | 0.17 | 0.18 | 0.18 | 0.18 | 0.19 |
| | Score-CAM | 0.05 | 0.06 | 0.08 | 0.09 | 0.10 | 0.11 | 0.13 | 0.14 | 0.15 | 0.16 | 0.16 | 0.17 | 0.18 | 0.18 | 0.19 |
| | Proposed | 0.04 | 0.05 | 0.07 | 0.08 | 0.09 | 0.09 | 0.10 | 0.11 | 0.12 | 0.14 | 0.15 | 0.17 | 0.18 | 0.19 | 0.19 |
| DR-Tra | Grad-CAM | 0.03 | 0.05 | 0.06 | 0.06 | 0.07 | 0.07 | 0.07 | 0.08 | 0.08 | 0.08 | 0.09 | 0.09 | 0.09 | 0.09 | 0.10 |
| | Grad-CAM++ | 0.04 | 0.06 | 0.07 | 0.07 | 0.08 | 0.08 | 0.09 | 0.09 | 0.10 | 0.10 | 0.10 | 0.11 | 0.11 | 0.11 | 0.11 |
| | Score-CAM | 0.04 | 0.05 | 0.06 | 0.06 | 0.07 | 0.08 | 0.08 | 0.09 | 0.09 | 0.10 | 0.11 | 0.11 | 0.11 | 0.12 | 0.12 |
| | Proposed | 0.03 | 0.04 | 0.05 | 0.05 | 0.06 | 0.06 | 0.06 | 0.07 | 0.07 | 0.07 | 0.08 | 0.08 | 0.08 | 0.09 | 0.10 |

Table A5: AOPC values on the MNIST dataset for different explanation methods across *CapsNet* variants over perturbation steps $b = 1$ to 15.

| CapsNet | Method | Perturbation Step $b$ | | | | | | | | | | | | | | |
|---|---|---|---|---|---|---|---|---|---|---|---|---|---|---|---|---|
| | | 1 | 2 | 3 | 4 | 5 | 6 | 7 | 8 | 9 | 10 | 11 | 12 | 13 | 14 | 15 |
| DR | Grad-CAM | 0.04 | 0.06 | 0.07 | 0.08 | 0.09 | 0.10 | 0.10 | 0.11 | 0.12 | 0.13 | 0.13 | 0.14 | 0.14 | 0.15 | 0.16 |
| | Grad-CAM++ | 0.04 | 0.06 | 0.07 | 0.08 | 0.09 | 0.10 | 0.11 | 0.11 | 0.12 | 0.13 | 0.14 | 0.15 | 0.15 | 0.16 | 0.16 |
| | Score-CAM | 0.05 | 0.06 | 0.07 | 0.08 | 0.09 | 0.09 | 0.10 | 0.11 | 0.11 | 0.12 | 0.12 | 0.13 | 0.13 | 0.14 | 0.14 |
| | Proposed | 0.05 | 0.06 | 0.07 | 0.08 | 0.09 | 0.10 | 0.11 | 0.11 | 0.12 | 0.12 | 0.13 | 0.14 | 0.15 | 0.16 | 0.17 |
| DR-Conv | Grad-CAM | 0.07 | 0.08 | 0.09 | 0.10 | 0.11 | 0.12 | 0.13 | 0.13 | 0.14 | 0.15 | 0.16 | 0.16 | 0.17 | 0.18 | 0.19 |
| | Grad-CAM++ | 0.07 | 0.08 | 0.09 | 0.10 | 0.11 | 0.12 | 0.13 | 0.13 | 0.14 | 0.15 | 0.16 | 0.16 | 0.17 | 0.18 | 0.19 |
| | Score-CAM | 0.08 | 0.09 | 0.10 | 0.11 | 0.12 | 0.13 | 0.14 | 0.15 | 0.16 | 0.16 | 0.17 | 0.18 | 0.19 | 0.19 | 0.19 |
| | Proposed | 0.07 | 0.07 | 0.09 | 0.11 | 0.11 | 0.12 | 0.13 | 0.14 | 0.14 | 0.15 | 0.16 | 0.17 | 0.17 | 0.18 | 0.18 |
| DR-Tra | Grad-CAM | 0.05 | 0.05 | 0.06 | 0.07 | 0.08 | 0.08 | 0.09 | 0.09 | 0.10 | 0.10 | 0.10 | 0.11 | 0.11 | 0.12 | 0.12 |
| | Grad-CAM++ | 0.04 | 0.05 | 0.06 | 0.06 | 0.07 | 0.07 | 0.08 | 0.08 | 0.08 | 0.09 | 0.09 | 0.10 | 0.10 | 0.11 | 0.11 |
| | Score-CAM | 0.06 | 0.07 | 0.08 | 0.09 | 0.10 | 0.11 | 0.11 | 0.12 | 0.13 | 0.13 | 0.14 | 0.15 | 0.15 | 0.16 | 0.16 |
| | Proposed | 0.06 | 0.07 | 0.08 | 0.09 | 0.10 | 0.11 | 0.11 | 0.12 | 0.13 | 0.13 | 0.14 | 0.15 | 0.16 | 0.16 | 0.17 |

Table A6: AOPC values on the SVHN dataset for different explanation methods across *CapsNet* variants over perturbation steps $b = 1$ to 15.

| CapsNet | Method | Perturbation Step $b$ | | | | | | | | | | | | | | |
|---|---|---|---|---|---|---|---|---|---|---|---|---|---|---|---|---|
| | | 1 | 2 | 3 | 4 | 5 | 6 | 7 | 8 | 9 | 10 | 11 | 12 | 13 | 14 | 15 |
| DR | Grad-CAM | 0.01 | 0.01 | 0.01 | 0.01 | 0.01 | 0.02 | 0.02 | 0.02 | 0.02 | 0.02 | 0.02 | 0.02 | 0.02 | 0.02 | 0.02 |
| | Grad-CAM++ | 0.01 | 0.02 | 0.02 | 0.02 | 0.02 | 0.02 | 0.02 | 0.02 | 0.03 | 0.03 | 0.03 | 0.03 | 0.03 | 0.03 | 0.03 |
| | Score-CAM | 0.01 | 0.01 | 0.01 | 0.01 | 0.01 | 0.01 | 0.01 | 0.02 | 0.02 | 0.02 | 0.02 | 0.02 | 0.02 | 0.02 | 0.02 |
| | Proposed | 0.01 | 0.01 | 0.01 | 0.01 | 0.02 | 0.02 | 0.02 | 0.02 | 0.02 | 0.02 | 0.02 | 0.02 | 0.02 | 0.02 | 0.03 |
| DR-Conv | Grad-CAM | 0.01 | 0.01 | 0.02 | 0.02 | 0.02 | 0.02 | 0.02 | 0.03 | 0.03 | 0.03 | 0.03 | 0.03 | 0.03 | 0.03 | 0.03 |
| | Grad-CAM++ | 0.01 | 0.01 | 0.01 | 0.01 | 0.01 | 0.01 | 0.02 | 0.02 | 0.02 | 0.02 | 0.02 | 0.02 | 0.02 | 0.02 | 0.02 |
| | Score-CAM | 0.01 | 0.01 | 0.01 | 0.01 | 0.01 | 0.02 | 0.02 | 0.02 | 0.02 | 0.02 | 0.02 | 0.02 | 0.02 | 0.02 | 0.02 |
| | Proposed | 0.01 | 0.01 | 0.01 | 0.02 | 0.02 | 0.02 | 0.02 | 0.02 | 0.02 | 0.02 | 0.02 | 0.02 | 0.02 | 0.03 | 0.03 |
| DR-Tra | Grad-CAM | 0.00 | 0.01 | 0.02 | 0.02 | 0.02 | 0.02 | 0.02 | 0.02 | 0.02 | 0.02 | 0.03 | 0.03 | 0.03 | 0.03 | 0.03 |
| | Grad-CAM++ | 0.01 | 0.01 | 0.02 | 0.02 | 0.02 | 0.02 | 0.02 | 0.02 | 0.02 | 0.03 | 0.03 | 0.03 | 0.03 | 0.03 | 0.03 |
| | Score-CAM | 0.01 | 0.01 | 0.01 | 0.01 | 0.01 | 0.02 | 0.02 | 0.02 | 0.02 | 0.02 | 0.02 | 0.02 | 0.02 | 0.03 | 0.03 |
| | Proposed | 0.01 | 0.01 | 0.01 | 0.01 | 0.02 | 0.02 | 0.02 | 0.02 | 0.02 | 0.02 | 0.02 | 0.02 | 0.02 | 0.03 | 0.03 |

Table A7: AOPC values on the CIFAR-10 dataset for different explanation methods across *CapsNet* variants over perturbation steps $b = 1$ to 15.

| CapsNet | Method | Perturbation Step $b$ | | | | | | | | | | | | | | |
|---|---|---|---|---|---|---|---|---|---|---|---|---|---|---|---|---|
| | | 1 | 2 | 3 | 4 | 5 | 6 | 7 | 8 | 9 | 10 | 11 | 12 | 13 | 14 | 15 |
| *DR* | Grad-CAM | 0.01 | 0.02 | 0.03 | 0.03 | 0.04 | 0.04 | 0.05 | 0.05 | 0.06 | 0.07 | 0.07 | 0.08 | 0.09 | 0.09 | 0.10 |
| | Grad-CAM++ | 0.02 | 0.03 | 0.04 | 0.04 | 0.05 | 0.06 | 0.06 | 0.07 | 0.08 | 0.08 | 0.09 | 0.09 | 0.10 | 0.10 | 0.11 |
| | Score-CAM | 0.01 | 0.02 | 0.02 | 0.03 | 0.03 | 0.04 | 0.04 | 0.05 | 0.05 | 0.06 | 0.06 | 0.06 | 0.06 | 0.07 | 0.07 |
| | Proposed | 0.02 | 0.03 | 0.04 | 0.05 | 0.06 | 0.07 | 0.08 | 0.09 | 0.09 | 0.10 | 0.11 | 0.12 | 0.12 | 0.13 | 0.14 |
| *DR-Conv* | Grad-CAM | 0.05 | 0.08 | 0.12 | 0.15 | 0.18 | 0.20 | 0.23 | 0.25 | 0.26 | 0.27 | 0.29 | 0.32 | 0.33 | 0.34 | 0.35 |
| | Grad-CAM++ | 0.03 | 0.05 | 0.07 | 0.10 | 0.11 | 0.13 | 0.16 | 0.18 | 0.20 | 0.22 | 0.24 | 0.26 | 0.28 | 0.30 | 0.32 |
| | Score-CAM | 0.05 | 0.08 | 0.09 | 0.11 | 0.14 | 0.17 | 0.19 | 0.21 | 0.23 | 0.26 | 0.27 | 0.30 | 0.31 | 0.33 | 0.35 |
| | Proposed | 0.04 | 0.09 | 0.11 | 0.13 | 0.16 | 0.18 | 0.20 | 0.22 | 0.25 | 0.27 | 0.29 | 0.31 | 0.32 | 0.34 | 0.36 |
| *DR-Tra* | Grad-CAM | 0.08 | 0.13 | 0.18 | 0.22 | 0.27 | 0.31 | 0.35 | 0.39 | 0.40 | 0.43 | 0.44 | 0.45 | 0.46 | 0.46 | 0.51 |
| | Grad-CAM++ | 0.09 | 0.15 | 0.20 | 0.25 | 0.29 | 0.30 | 0.33 | 0.35 | 0.37 | 0.38 | 0.40 | 0.43 | 0.44 | 0.46 | 0.50 |
| | Score-CAM | 0.03 | 0.07 | 0.10 | 0.12 | 0.15 | 0.17 | 0.20 | 0.22 | 0.25 | 0.27 | 0.30 | 0.32 | 0.35 | 0.38 | 0.40 |
| | Proposed | 0.05 | 0.12 | 0.15 | 0.18 | 0.21 | 0.24 | 0.27 | 0.29 | 0.32 | 0.35 | 0.39 | 0.42 | 0.45 | 0.48 | 0.51 |

Table A8: AOPC values on the MSTAR dataset for different explanation methods across *CapsNet* variants over perturbation steps $b = 1$ to 15.

| Dataset | Method | $k=1$ | 2 | 3 | 4 | 5 | 6 | 7 | 8 | 9 | 10 | 11 | 12 | 13 | 14 | 15 |
|---|---|---|---|---|---|---|---|---|---|---|---|---|---|---|---|---|
| MSTAR | Grad-CAM | 2.57 | 2.91 | 3.17 | 3.31 | 3.45 | 3.47 | 3.54 | 3.60 | 3.70 | 3.76 | 3.78 | 3.78 | 3.79 | 3.87 | 3.87 |
| | Grad-CAM++ | 2.68 | 2.96 | 3.24 | 3.35 | 3.44 | 3.48 | 3.56 | 3.59 | 3.61 | 3.68 | 3.75 | 3.76 | 3.76 | 3.82 | 3.83 |
| | Score-CAM | 2.07 | 2.40 | 2.94 | 3.22 | 3.37 | 3.46 | 3.56 | 3.66 | 3.73 | 3.78 | 3.82 | 3.82 | 3.85 | 3.86 | 3.86 |
| | Proposed | 2.72 | 3.03 | 3.25 | 3.30 | 3.31 | 3.37 | 3.49 | 3.54 | 3.57 | 3.57 | 3.65 | 3.65 | 3.67 | 3.70 | 3.92 |
| MNIST | Grad-CAM | 0.40 | 0.54 | 0.62 | 0.69 | 0.82 | 0.90 | 0.95 | 0.99 | 1.07 | 1.08 | 1.11 | 1.12 | 1.14 | 1.16 | 1.19 |
| | Grad-CAM++ | 0.34 | 0.45 | 0.49 | 0.55 | 0.66 | 0.73 | 0.77 | 0.80 | 0.88 | 0.91 | 0.96 | 0.99 | 1.04 | 1.08 | 1.11 |
| | Score-CAM | 0.46 | 0.61 | 0.70 | 0.78 | 0.92 | 1.02 | 1.09 | 1.14 | 1.21 | 1.31 | 1.35 | 1.41 | 1.47 | 1.53 | 1.58 |
| | Proposed | 0.32 | 0.43 | 0.47 | 0.54 | 0.64 | 0.74 | 0.78 | 0.86 | 0.92 | 0.95 | 0.99 | 1.06 | 1.13 | 1.17 | 1.23 |
| SVHN | Grad-CAM | 0.33 | 0.38 | 0.41 | 0.44 | 0.46 | 0.47 | 0.49 | 0.49 | 0.50 | 0.51 | 0.53 | 0.55 | 0.57 | 0.58 | 0.59 |
| | Grad-CAM++ | 0.32 | 0.37 | 0.39 | 0.43 | 0.44 | 0.45 | 0.47 | 0.49 | 0.50 | 0.51 | 0.53 | 0.54 | 0.56 | 0.57 | 0.58 |
| | Score-CAM | 0.33 | 0.37 | 0.38 | 0.41 | 0.43 | 0.43 | 0.45 | 0.47 | 0.48 | 0.49 | 0.51 | 0.53 | 0.57 | 0.58 | 0.59 |
| | Proposed | 0.30 | 0.32 | 0.37 | 0.41 | 0.42 | 0.44 | 0.46 | 0.47 | 0.50 | 0.51 | 0.53 | 0.55 | 0.57 | 0.59 | 0.59 |
| CIFAR-10 | Grad-CAM | 0.07 | 0.07 | 0.09 | 0.11 | 0.13 | 0.12 | 0.15 | 0.15 | 0.17 | 0.18 | 0.18 | 0.18 | 0.20 | 0.19 | 0.22 |
| | Grad-CAM++ | 0.06 | 0.07 | 0.08 | 0.11 | 0.11 | 0.10 | 0.12 | 0.11 | 0.13 | 0.13 | 0.15 | 0.17 | 0.17 | 0.18 | 0.18 |
| | Score-CAM | 0.05 | 0.05 | 0.08 | 0.10 | 0.11 | 0.13 | 0.12 | 0.13 | 0.14 | 0.15 | 0.16 | 0.18 | 0.18 | 0.19 | 0.20 |
| | Proposed | 0.10 | 0.11 | 0.13 | 0.15 | 0.17 | 0.18 | 0.19 | 0.21 | 0.22 | 0.23 | 0.23 | 0.23 | 0.25 | 0.26 | 0.28 |

Table A9: AOPC results using EM Routing (EMR) across datasets

| Dataset | MNIST | | | | SVHN | | | | CIFAR-10 | | | | MSTAR | | | |
|---|---|---|---|---|---|---|---|---|---|---|---|---|---|---|---|---|
| Method | DR | *DR-Conv* | *DR-Tra* | *EM* | DR | *DR-Conv* | *DR-Tra* | *EM* | DR | *DR-Conv* | *DR-Tra* | *EM* | DR | *DR-Conv* | *DR-Tra* | *EM* |
| **$Z = 2$** | | | | | | | | | | | | | | | | |
| Caps-CAM (Proposed) | 0.2 | 0.6 | 14 | 46 | 3 | 9 | 9 | 31 | 14 | 17 | 15 | 35 | 3 | 1 | 21 | 37 |
| **$Z = 4$** | | | | | | | | | | | | | | | | |
| Caps-CAM (Proposed) | 0.1 | 1 | 12 | 83 | 3 | 2 | 9 | 34 | 13 | 17 | 15 | 27 | 3 | 1 | 18 | 50 |
| **$Z = 6$** | | | | | | | | | | | | | | | | |
| Caps-CAM (Proposed) | 0.2 | 1 | 13 | 84 | 3 | 3 | 9 | 34 | 14 | 17 | 15 | 26 | 3 | 1 | 19 | 50 |
| **$Z = 8$** | | | | | | | | | | | | | | | | |
| Caps-CAM (Proposed) | 0.2 | 1 | 12 | 86 | 3 | 2 | 9 | 35 | 14 | 14 | 15 | 26 | 3 | 1 | 20 | 47 |

Table A10: Average % drop (lower is better) for the proposed Caps-CAM explanation method across MNIST, SVHN, CIFAR-10, and MSTAR datasets. Models compared are Dynamic Routing (*DR*), DR with deep convolutional layers (*DR-Conv*), DR with transformer blocks before the *PC* layer (*DR-Tra*), and EM-routing (*EM*). Results are shown for different values of $Z$, as $Z$ is being the number of capsule dimensions.

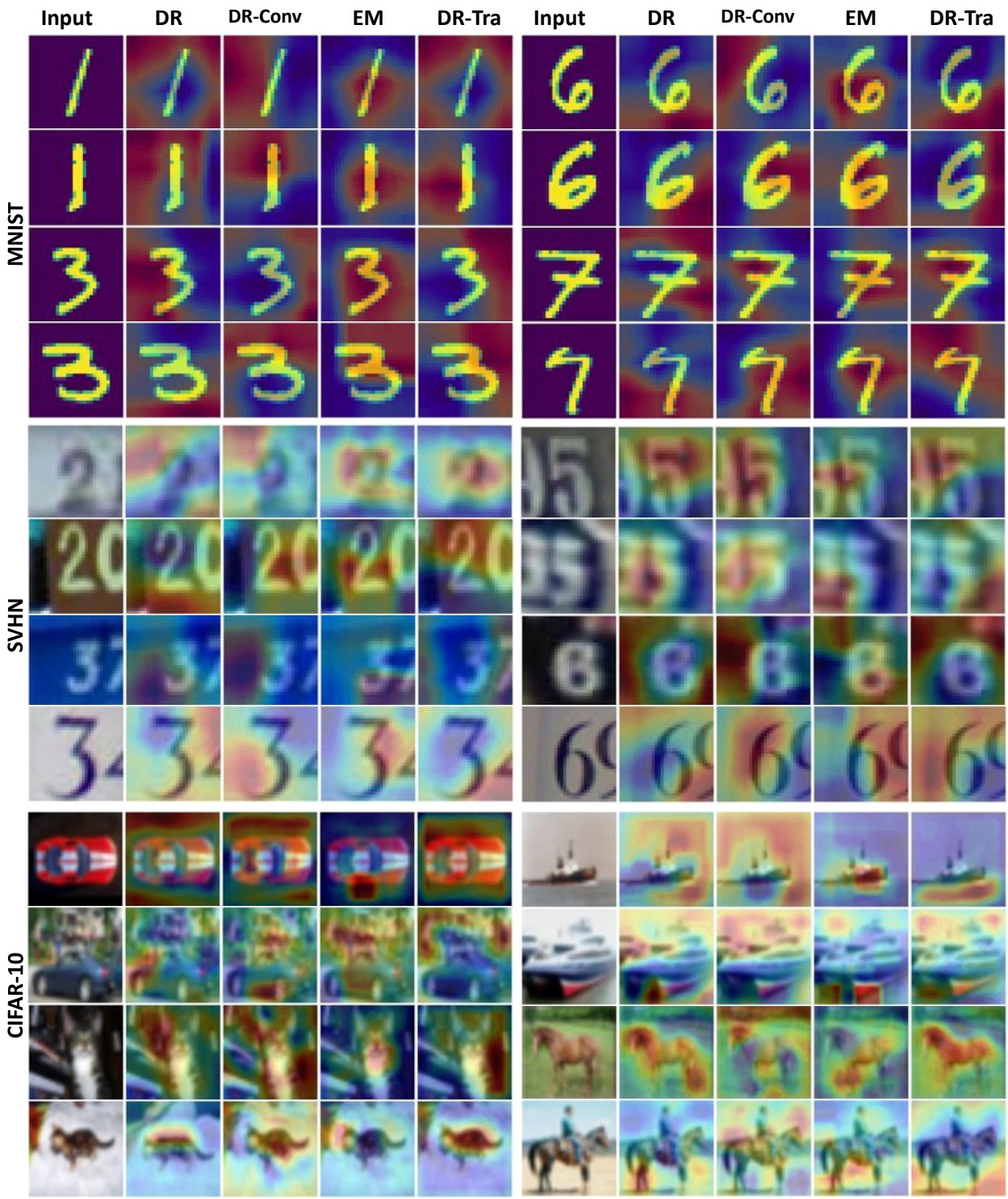

Figure A3: Additional heatmaps produced by *Caps-CAM* method across the used datasets.

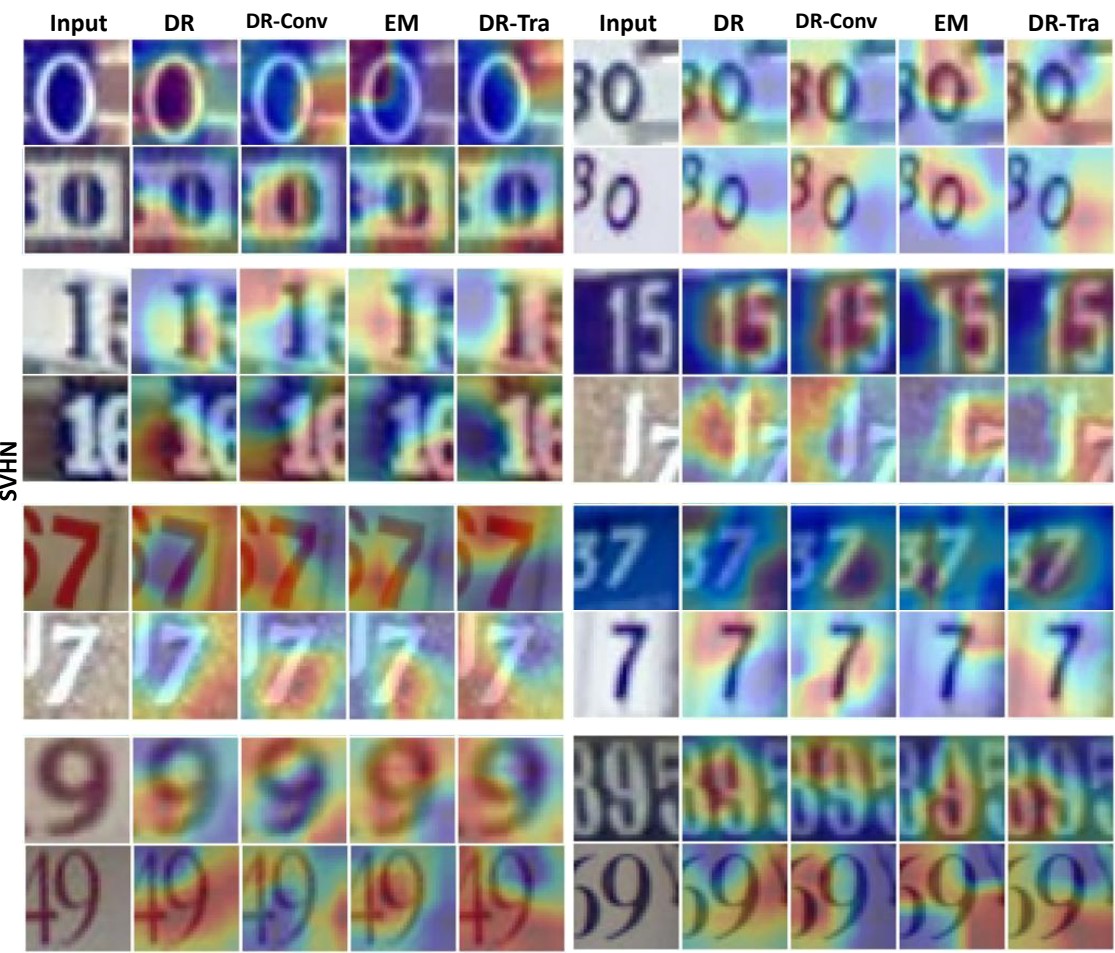

Figure A4: Additional heatmaps generated by the proposed *Caps-CAM* method for classes in the SVHN dataset.

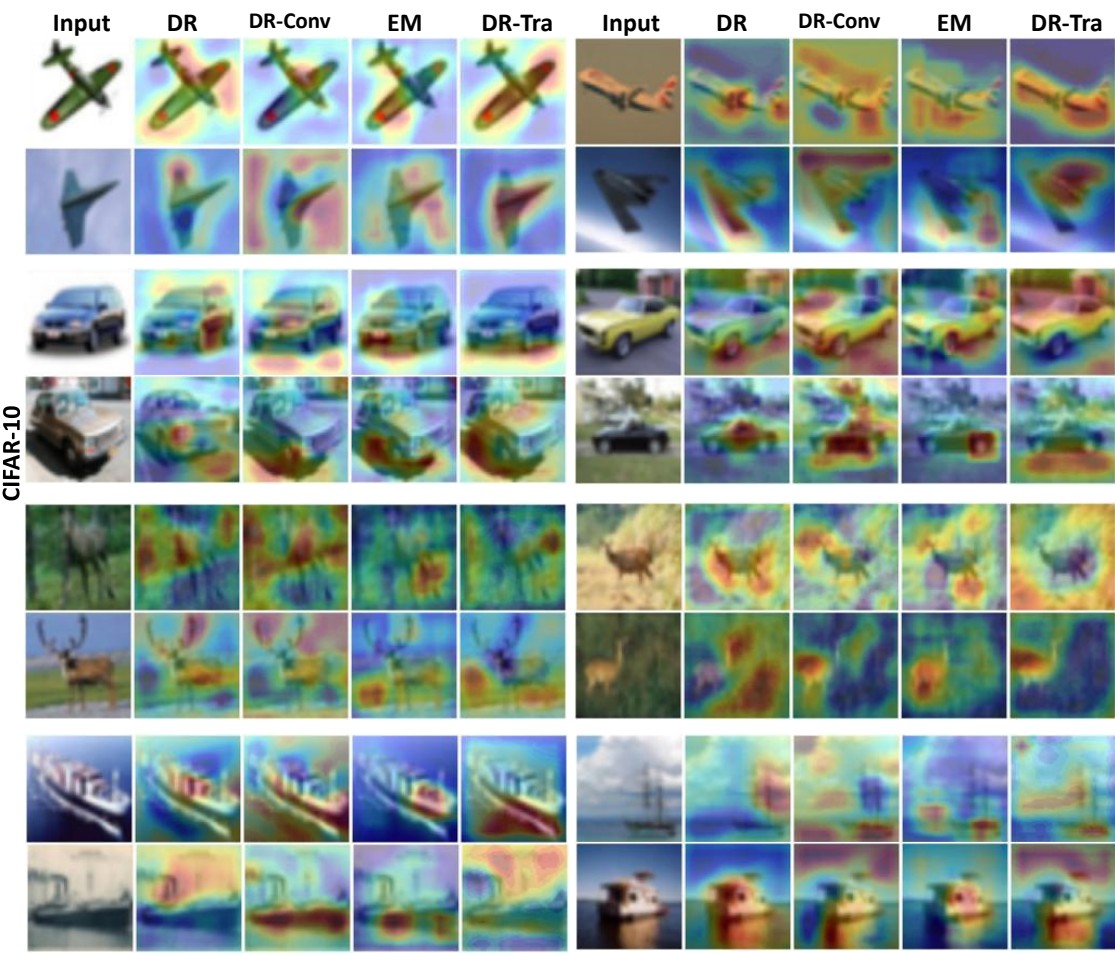

Figure A5: Additional heatmaps generated by the proposed *Caps-CAM* method for classes in the CIFAR-10 dataset.

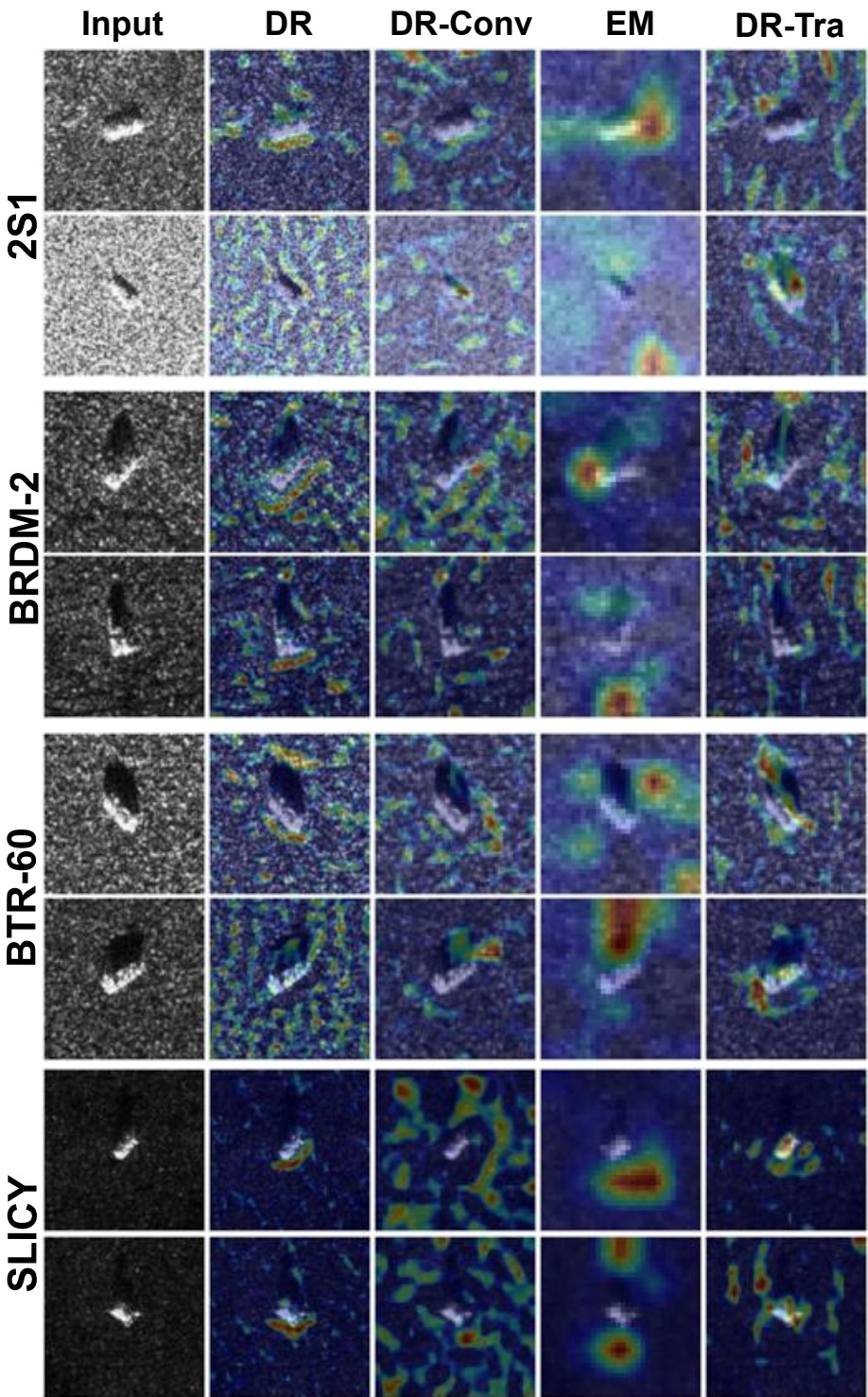

Figure A6: Additional heatmaps generated by the proposed *Caps-CAM* method for heavy tank classes in the MSTAR dataset.

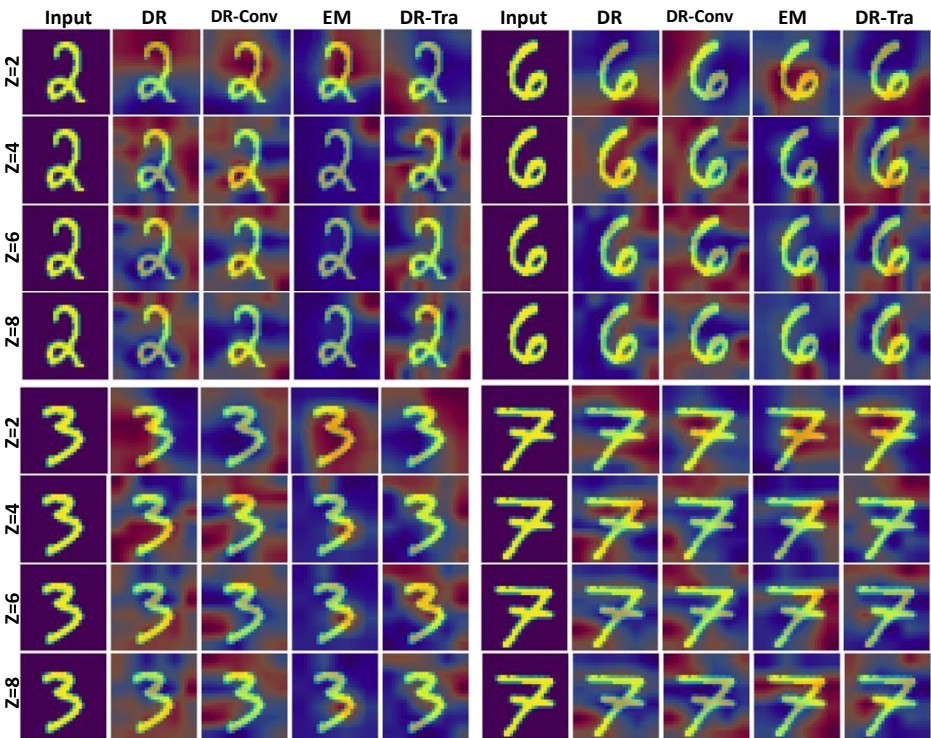

Figure A7: Visual heatmap of the selected top capsule dimensions $Z$ on MNIST dataset

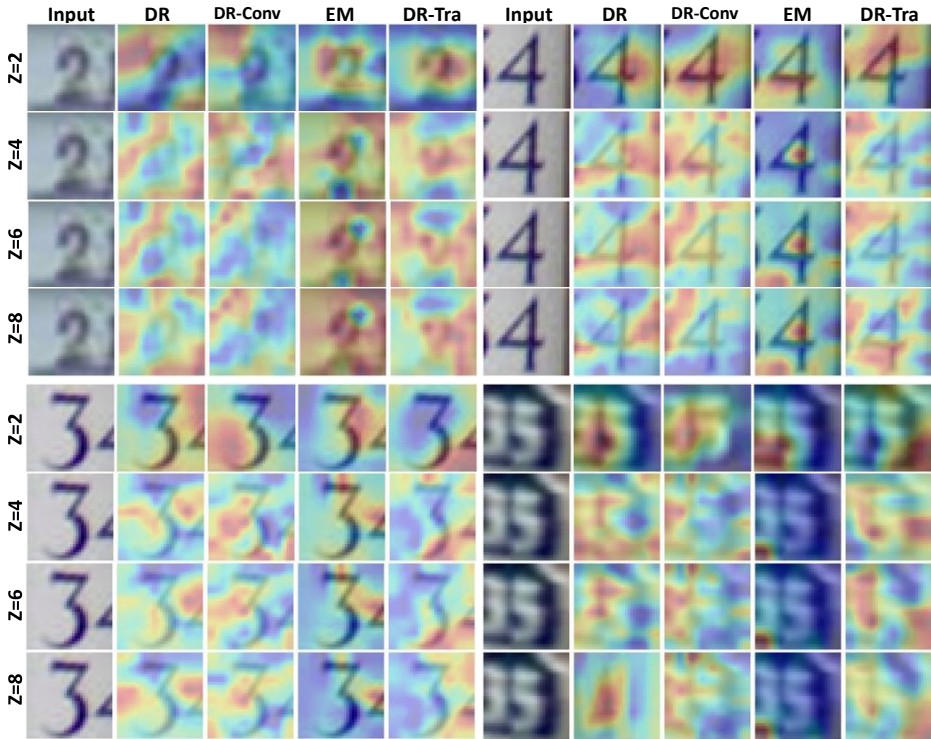

Figure A8: Visual heatmap of the selected top capsule dimensions $Z$ on SVHN dataset

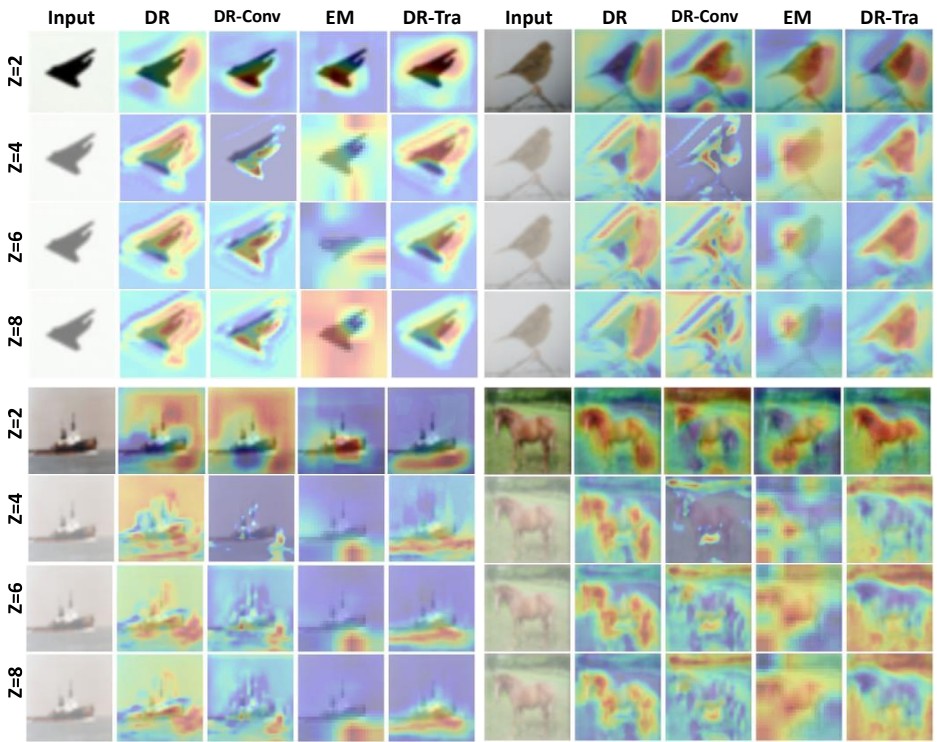

Figure A9: Visual heatmap of the selected top capsule dimensions $Z$ on CIFAR-10 dataset

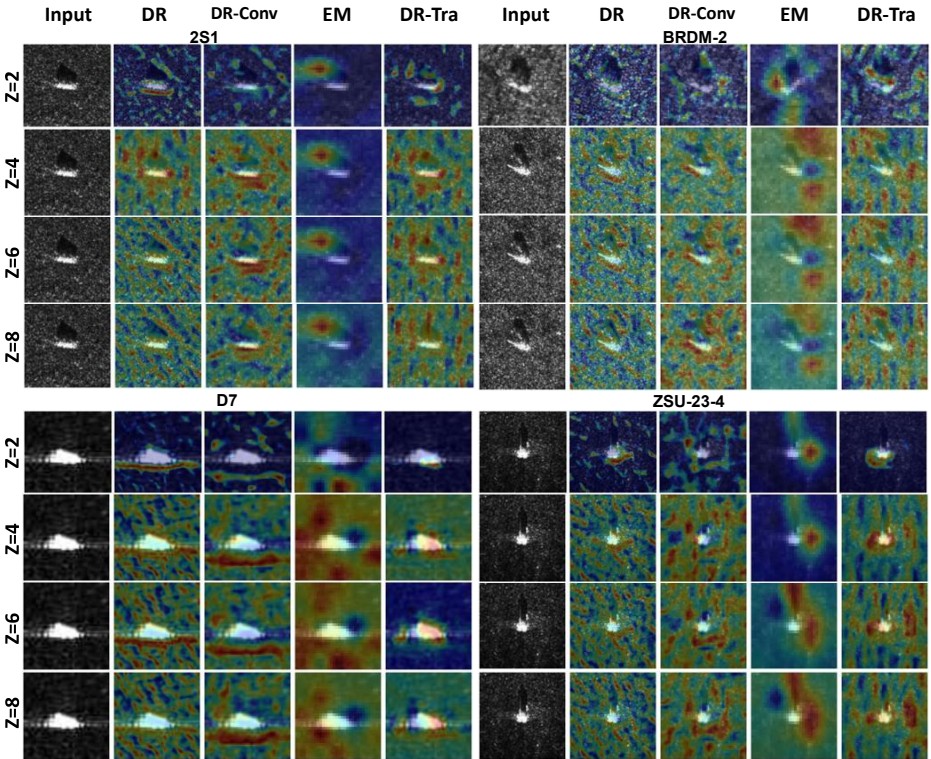

Figure A10: Visual heatmap of the selected top capsule dimensions $Z$ on MSTAR dataset

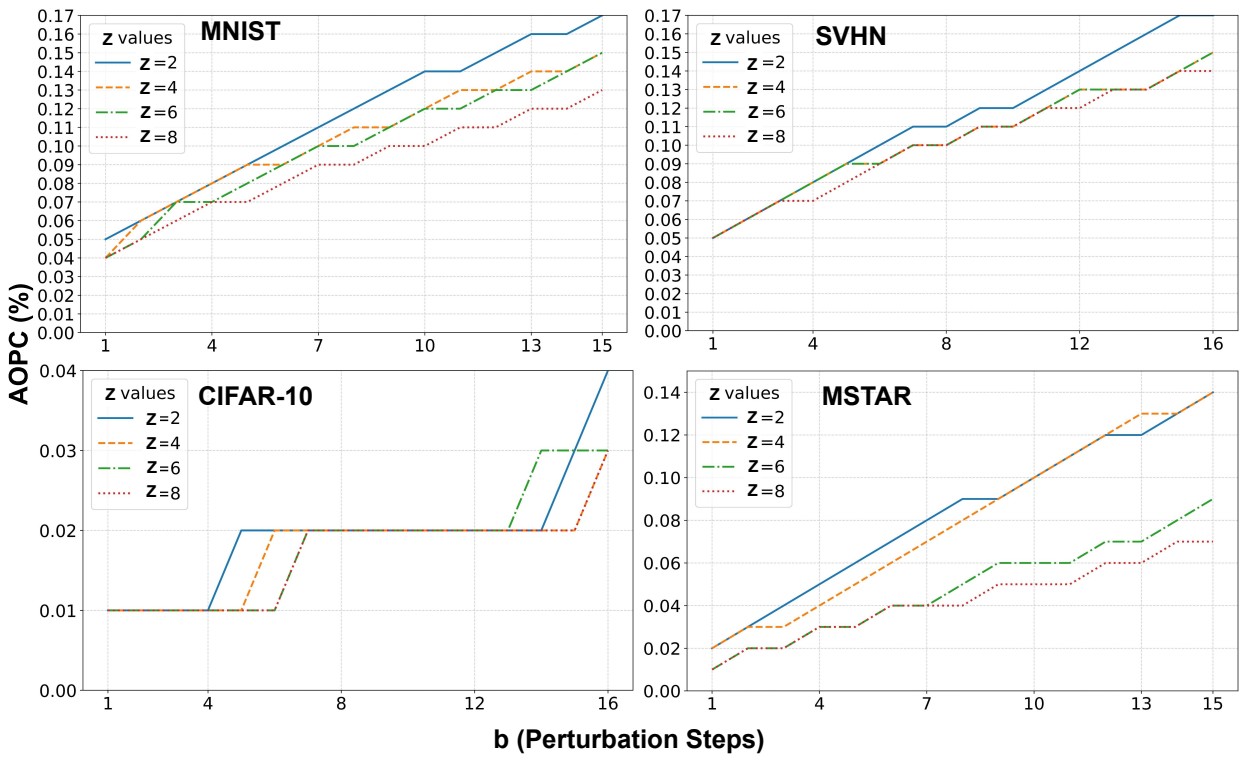

Figure A11: Effect of top capsule dimension $Z$ selection on $AOPC$ under $DR$ variant

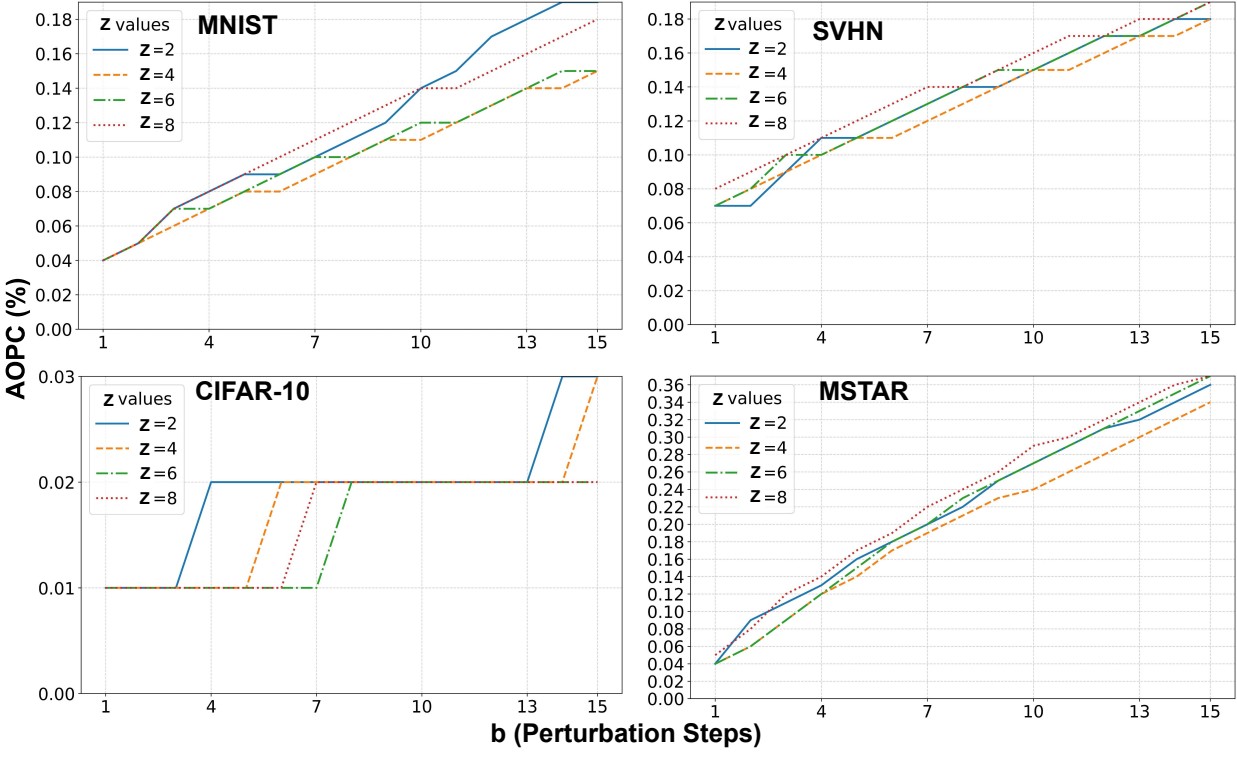

Figure A12: Effect of top capsule dimension $Z$ selection on $AOPC$ under $DR\text{-}CNNs$ variant

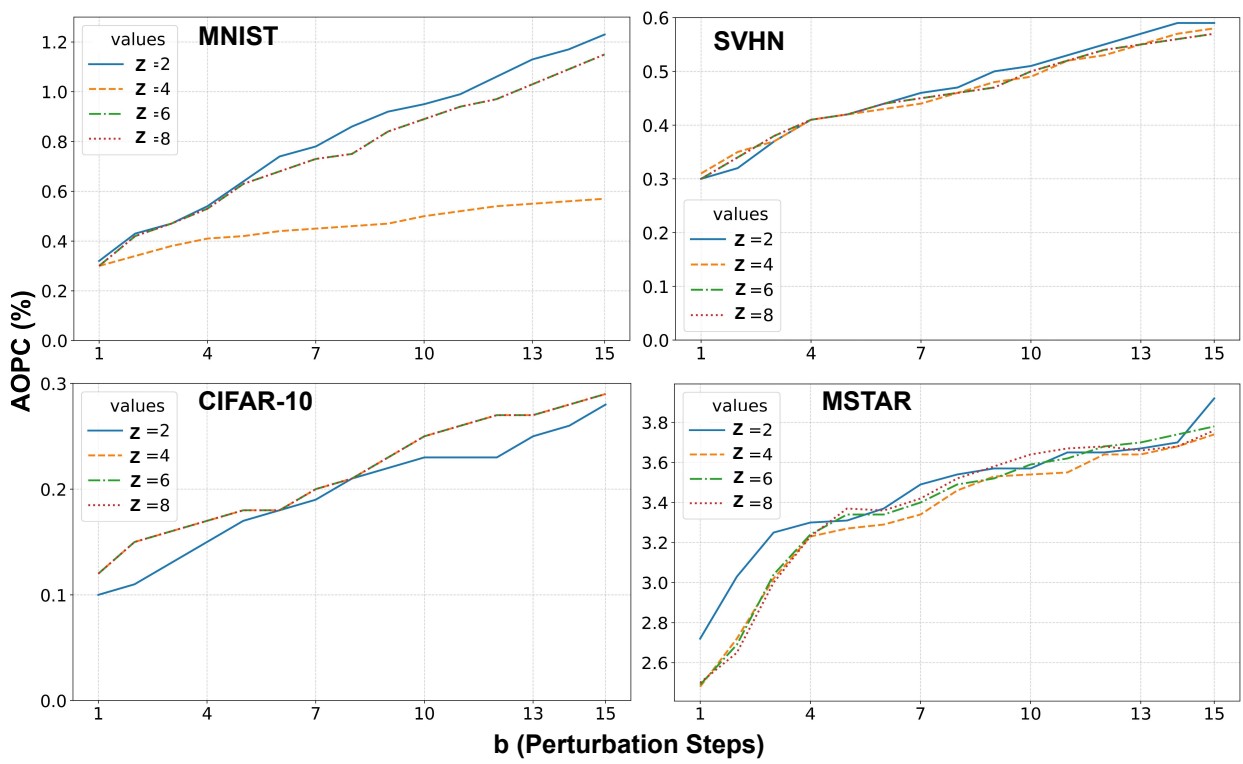

Figure A13: Effect of top capsule dimension $Z$ selection on $AOPC$ under $EMR$ variant

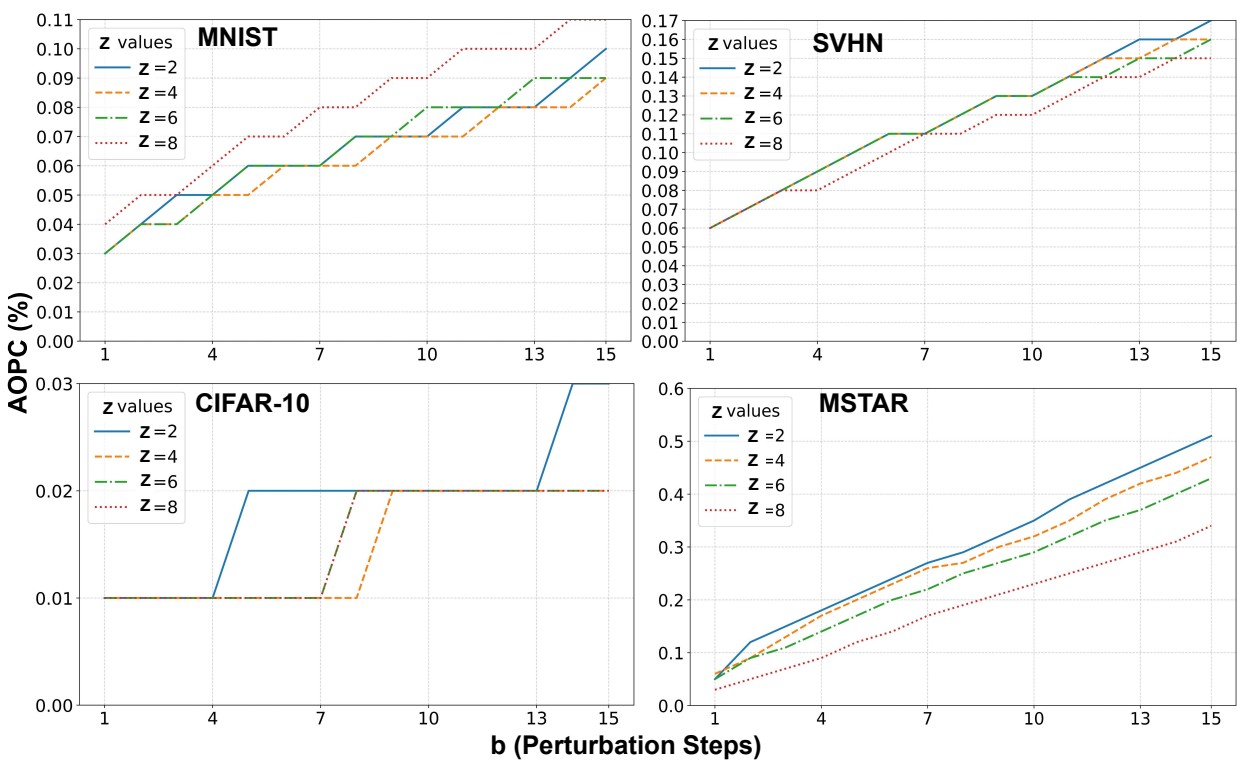

Figure A14: Effect of top capsule dimension $Z$ selection on $AOPC$ under $DR$-$Tra$ variant