# OpenReview forum: "Visual Explanations for Capsule Networks"
_TMLR — Accepted by TMLR_

### Review · Reviewer_H3nP · 2026-03-23

**Summary Of Contributions:**

The paper proposes Caps-CAM, a new post-hoc and local explainability technique for Capsule Networks. The key difference from previous approaches is that in Caps-CAM the capsule activation map is weighted based on its corresponding gradient therefore accounting for the relevance of each capsule to each class. Specifically gradient information is used to measure the relevance of capsules and retaining only the top-ranking ones before computing their contributions.
Quantitative experimental results across three benchmark datasets (MNIST, SVHN, and CIFAR-10) and one additional use case dataset (MSTAR) show that the proposed method outperforms the baselines (Grad-CAM, Grad-CAM++, and Score-CAM) in most examined settings in terms of Average % drop. In terms of AOPC, the method is competitive, but sometimes lose to Score-CAM (on benchmark datasets) and Grad-CAM (on MSTAR).

**Audience:**

Yes

**Audience Explanation:**

Given the popularity of both Capsule Networks and Class Activation Mapping (CAM) based explainability techniques, I think this paper would be of interest to the TMLR audience.

**Claims And Evidence:**

No

**Claims Explanation:**

- While reading the paper I found the separate roles of Section 3.2 and Section 3.3 not entirely clear. Continuing reading the paper, it seems that Section 3.2 represents a naive adaptation of Score-CAM (and similar) to CapsNets while 3.3 is the proposed approach. But it is not entirely clear when I first read it and it would be better to indicate this.

- I found the description of the methodology a bit hard to follow and I think it would benefit from a short pseudo code. In addition the text could be polished for clarity. A couple of examples:
	* “In this step, … are weighted using the corresponding gradients to identify …”. However so far there was no mention of gradients. Only later we have “Towards this goal, we first reintroduce the use of gradients …”
	* “To achieve this, we reshape the activations ….” however the sentence does not indicate what is it reshaped to.
	* Eq. (4) is \forall k but k is not presented in the notation of $R_i$. Should this be $R_i^k$?

- The abstract indicates various domains including “legal document analysis”, however the proposed approach seems largely focused on images and the experiments are limited to images. It is not clear whether this approach is applicable and/or effective for other domains like legal document analysis that is specifically mentioned in the abstract.

- The quantitative analysis uses two metrics however omits some popular metrics. In particular, the increase-in-confidence and Win % were introduced as a complementary metrics alongside the average drop % in [Chattopadhay et al., 2018] but are missing in the paper, as well as the insertion and deletion metrics [1]. Extending the experimental evaluation with additional widely used metrics can strengthen the support for the claim that the proposed approach outperforms the baselines.

- The paper seems to focus on the top 2 capsules based on relevance. It is not clear why 2 was selected, whether this is proposed as a general value for all datasets, or should be tuned per dataset. Also there is no sensitivity analysis to show the impact of this choice.

[1] Petsiuk, V., Das, A., & Saenko, K. (2018). Rise: Randomized input sampling for explanation of black-box models. arXiv preprint arXiv:1806.07421.

**Requested Changes:**

- Clarify/improve the structure of Section 3, in particular the role of Section 3.2 vs 3.3 (critical)
- Clarify the technical details in the methodology and add a pseudo-code (critical)
- Clarify whether the proposed approach extends beyond images as the abstract suggests (critical)
- Incorporate additional evaluation metrics, namely increase in confidence and win % (critical), as well as insertion and deletion (recommended)
- Clarify the choice of the top 2 capsules (critical) and perform sensitivity analysis (recommended)

---

> ### Author Response · Authors · 2026-05-02
> **Response to Reviewer H3nP**
>
> We thank the reviewer for the careful reading of our manuscript and for the constructive feedback. We appreciate the recognition that the topic is relevant to the TMLR audience, and we agree that the clarity of the methodology and the experimental evaluation can be improved. Below, we address each comment separately and describe the changes that will be incorporated in the revised version. Please note that all the changes can be found in blue in the revised manuscript.
>
> 1. We thank the reviewer for highlighting the need to clarify the structure of Sec. 3. Therefore, we clarify the roles of these sections and their relationship as follows.
> - Section 3.2 (Class Activation Mapping for CapsNets) introduces a baseline adaptation of Score-CAM to capsule networks (CapsNets). Specifically, it addresses the structural mismatch between CNN feature maps and capsule representations by (i) reshaping activations to expose the capsule dimension, (ii) aggregating information across capsule dimensions, and (iii) introducing capsule-aware normalization. This section establishes how the standard Score-CAM method can be applied to CapsNets without modifying their core behavior. However, this adaptation leads to information loss. In CapsNets, each capsule encodes both the presence and instantiation parameters of an entity through a vector representation, where the orientation of the vector carries meaningful information. To apply Score-CAM, these capsule dimensions must be reduced to scalar activation maps (e.g., via aggregation across dimensions). This reduction discards the directional information within capsule vectors and ignores part–whole relationships encoded through dynamic routing. As a result, the generated explanations do not fully reflect the structured representations learned by the network.
> - Section 3.3 (Caps-CAM) builds directly on this formulation (Section 3.2) and introduces our main contribution. In particular, it extends the baseline by incorporating gradient information to estimate capsule relevance, followed by a top-capsule selection. This allows the method to focus only on the most informative capsule dimensions, which improves the quality the resulting visual explanations.
> - In the revised version of the paper, we have added an explicit paragraph at the beginning of Sections 3.2 and 3.3 to clarify their relationship and to clearly distinguish their respective roles. In particular, we state that Sec 3.3 builds upon the foundations introduced in Section 3.2. We have also revised the transition between the two sections to emphasize that Section 3.2 provides the necessary background and formulation, while Section 3.3 presents the proposed method. Furthermore, we have included Algorithm 1, which summarizes the complete Caps-CAM pipeline and further clarifies how the two sections are connected.
>
> 2. We thank the reviewer for this important suggestion. In the revised manuscript, we have included a step-by-step pseudo-code describing the Caps-CAM procedure (Algorithm 1, Section 3.3). We have also revised the corresponding section to further clarify the method and address this concern. In addition, we will make the code publicly available upon acceptance to enhance the reproducibility of our method. We believe that these additions improve both the clarity of the method and the overall readability of the paper.
>
> 3. We thank the reviewer for this observation. Caps-CAM is designed for Capsule Networks (CapsNets) in general and is not inherently restricted to image data. However, in the current manuscript, the experimental evaluation focuses on vision datasets, as most existing CapsNet architectures and commonly used explanation metrics (e.g., CAM-based metrics) are primarily established in the visual domain.
> - To avoid over-claiming, we revised the manuscript to better clarify the scope of our contribution. In particular, we updated the abstract to state that the proposed method is applicable to domains where CapsNets are employed, including vision and other structured data, rather than implying a specific application domain. We also removed domain-specific examples, such as legal document analysis, that may suggest broader empirical validation than what is currently provided.
> - Nevertheless, we believe that the proposed method can be extended to non-visual domains with minimal modifications, primarily related to the nature of the input data.

---

> > ### Author Response · Authors · 2026-05-02
> > **Response to Reviewer H3nP**
> >
> > 4. We thank the reviewer for this valuable suggestion. We agree that incorporating additional evaluation metrics would further strengthen the empirical analysis of the proposed method. In the revised manuscript, we included the suggested Increase in Confidence. This metric will complement the existing AOPC and Average % Drop measures, which enable a more thorough assessment of the faithfulness and robustness of the generated explanations. Due to the computational cost associated with these experiments, the results are currently being generated. We will include as many results as possible, along with the corresponding analysis, in the revised version. Any remaining results will be incorporated into the final version to ensure a consistent evaluation.
> > - Regarding the Win % metrics, we kindly ask if you could provide a clear point to this metric?
> > - Regarding the insertion and deletion metric, while we acknowledge their importance, due to their high computational costs and our reduced computational resources, we delegate the evaluation using these metrics to be part of future work.
> >
> > 5. We thank the reviewer for this important comment. The reason for selecting only 2 capsule dimensions out of the 8 dimensions is that, in CapsNets, each capsule dimension encodes a distinct characteristic, whose relevance can be quantified via its activation strength. Focusing on the entire capsule dimensions can produce diffuse or overlapping activations, which may weaken the resulting explanation map. Considering only these specific capsule dimensions will help focus on the regions that are genuinely relevant when evaluating the generated explanation heatmaps.
> >  In Sec 4 (Explanation Methods), we briefly explained the choice of the number of relevant capsules in the first version of the paper.
> > - To further support this design choice (in the first version of the paper - Supplementary Material and we moved this evidence to the main paper) , we conducted an ablation analysis to study the effect of varying the number of selected capsules. This analysis is provided in the Supplementary Material, where we gradually increase the number of selected capsules and evaluate both: the qualitative quality of the generated visual heatmaps, and the quantitative evaluation metrics.
> > - Based on the results, selecting a small number of capsules (i.e., Z=2) achieves the highest or competitive performance across datasets and architectures, which indicates that the most relevant information driving the model’s prediction is concentrated in a small subset of capsule dimensions and that the top-2 capsules are sufficient to capture this core evidence. Furthermore, increasing Z does not lead to significant improvements and, in some cases, slightly degrades performance, which suggests that additional capsules tend to encode complementary or less discriminative information. At the same time, the results remain stable across different values of Z, demonstrating that the proposed method is not overly sensitive to this parameter.
> > - To address this concern, we revised Sec 4 in the main manuscript and moved some of the produced results experiments from the Supplementary Material to the main paper. We also added a discussion to explain the findings of the top capsules selections and evaluate them. This change will allow us to present the results together with the related discussion, which makes the explanation more explicit and easier to follow. We believe that including these experiments in the main text will improve the clarity of the presentation and better support the claims made in this section.

---

> ### Author Response · Authors · 2026-05-24
> **Response to Reviewer H3nP**
>
> o	We agree that quantitative validation is essential for establishing the reliability of explainability methods. The evaluation metrics adopted in this work are widely used and well-established in the explainability literature for evaluating the quality and faithfulness of visual explanations. To further strengthen the empirical analysis, we expanded the quantitative evaluation by introducing an additional confidence increase analysis, which measures the effect of the highlighted regions on the model’s prediction confidence. The corresponding results have been added to Table 3, together with an extended discussion in Section 4.2.
>
> 1. Across all architectures and datasets, Caps-CAM consistently achieves competitive or superior performance in terms of average % increase in confidence, particularly for the DR and DR-Conv models, where it ranks best overall. While a few isolated cases (e.g., DR-Tra on MNIST and EM on CIFAR-10) show slightly lower gains compared to alternatives, Caps-CAM remains generally strong across settings. Together with the average % drop results, these findings indicate that the regions identified by Caps-CAM are both sufficient to sustain predictions and highly influential when emphasized, reinforcing the method’s ability to reliably capture the most discriminative features used by CapsNet models.

---

### Review · Reviewer_77p8 · 2026-03-24

**Summary Of Contributions:**

This paper proposes Caps-CAM, a visual explanation method for feed-forward Capsule Networks. The method adapts the CAM/Score-CAM paradigm to capsule architectures by explicitly using capsule activations and gradients to score capsule relevance, selecting the top-relevant capsule dimensions, and then generating an attribution map through a Score-CAM-like masking pipeline. The main motivation is that prior CapsNet explanation methods are either tied to specific routing algorithms or are direct adaptations of CNN-oriented explanation methods that may ignore capsule structure. Empirically, the paper evaluates Caps-CAM on four CapsNet variants (DR, DR-Conv, DR-Tra, EM) across MNIST, SVHN, CIFAR-10, and an MSTAR use case, using qualitative heatmaps plus Average Drop and AOPC as quantitative evaluation metrics.

**The main strengths are:** the problem is well motivated; the method is reasonably intuitive and is designed around capsule-specific structure rather than treating the model as a standard CNN; the evaluation spans multiple capsule backbones and includes both toy benchmarks and a more application-oriented dataset; and the quantitative results are often favorable relative to the adapted Grad-CAM, Grad-CAM++, or Score-CAM baselines.

**The main weaknesses are:** the experimental comparison does not include the most relevant CapsNet-specific prior explanation baselines discussed in the related work; several claims are broader than the evidence currently supports (especially "generality", "superiority", and reduced computational overhead); some important design choices, notably the top-2 capsule selection, are justified mainly via supplementary ablations rather than in the main paper; and the method appears materially weaker for some EM cases, which deserves deeper analysis in the main text.

**Additional Comments:**

I found the paper easy to follow at a high level, and I think the problem it addresses is legitimate. My main concern is that the current version overstates what has been established. With a tighter empirical positioning, stronger baseline comparisons, and clearer discussion of limitations and failure cases, this could become a solid niche contribution on explainability for capsule architectures.

**Audience:**

Yes

**Audience Explanation:**

Yes. Even though Capsule Networks are a specialized topic, there is a real audience in TMLR interested in interpretability methods for non-standard neural architectures, especially when the method is architecture-aware rather than a direct transplant from CNN explainability. The paper also speaks to a broader theme in XAI: whether explanation methods should reflect the internal structure of the model being explained. That is a question with relevance beyond CapsNets themselves.

**Broader Impact Concerns:**

I do not see major broader-impact concerns that are unique to this paper.

**Claims And Evidence:**

No

**Claims Explanation:**

The paper does provide meaningful evidence for a narrower claim: namely, that Caps-CAM is a plausible and often effective explanation method for several feed-forward CapsNet image classifiers, and that it often compares favorably against adapted CNN-style CAM baselines under the chosen perturbation metrics. The qualitative figures are reasonably consistent with that story, and Table 2 shows strong Average % Drop results in many configurations, especially for DR-based models and on MSTAR outside the EM case.

However, the current evidence is not fully convincing for some of the broader claims made in the paper. First, the paper frames itself partly against prior CapsNet-specific explanation methods that rely on routing, but the empirical comparison is only against adapted Grad-CAM, Grad-CAM++, and Score-CAM. Without direct comparison to the most relevant capsule-specific prior methods discussed in Section 2, it is difficult to assess whether Caps-CAM is truly the strongest available explanation approach for CapsNets rather than simply stronger than generic CNN-derived baselines.

Second, the paper claims architectural generality and reduced computational overhead, but the evidence for both is incomplete. The method is evaluated on four image-classification CapsNet variants, which is useful, but still narrower than the phrase "general method for feed-forward CapsNet architectures" suggests. Likewise, the computational-overhead claim is asserted in the introduction and method discussion, yet I did not see a direct runtime, memory, or forward-pass-count comparison substantiating that point.

Third, the quantitative picture is positive but not uniformly strong, and the paper itself notes important exceptions. Table 2 shows that Caps-CAM underperforms Score-CAM for at least the EM model on MNIST and MSTAR, and is slightly worse on DR-Tra for CIFAR-10. The paper acknowledges some of these cases, which I appreciate, but these failures weaken a blanket "superiority" claim and suggest the method may be more architecture-dependent than the presentation implies.

Finally, some central design choices remain under-justified in the main paper. In particular, the use of the top-2 capsules seems important to the method’s behavior, but the supporting ablation is deferred to the supplementary material. Similarly, the handling of EM-routing appears to require special treatment and still yields weaker results in some settings, which should be analyzed more directly in the main text if the authors want to argue that the method is broadly robust across capsule architectures.

**Requested Changes:**

**I think these changes are critical:**

1. Add direct empirical comparison to the most relevant prior CapsNet-specific explanation methods discussed in Section 2, not only to adapted CNN-based CAM methods. If some baselines are genuinely incompatible with certain architectures, that should be explained carefully, and the empirical comparison should still cover the settings where they are applicable. This is the main gap preventing me from viewing the current evidence as fully convincing.

2. Narrow or better support the paper’s strongest claims. In particular, I would revise wording around "general method", "superiority", and "reduced computational overhead" unless the authors add evidence that directly supports those statements. At present, the experiments demonstrate effectiveness on several feed-forward CapsNet image classifiers, which is a meaningful result, but still narrower than the current framing. A simple runtime complexity comparison would help with the overhead claim.

3. Clarify the method description around Eqs. (4)-(6). The paper should make more explicit how the relevance scores are computed and ranked, how the sign/magnitude of the gradient enters, exactly how the top capsules are selected, and how the EM case differs in practice. A short algorithm block or pseudocode would substantially improve clarity and reproducibility.

4. Bring the key ablation on the number of selected capsule dimensions into the main paper, or at least summarize it there. Since the choice of top-2 capsules is central to the proposed method and to its claimed denoising effect, the reader should not have to rely on supplementary figures to understand whether this design choice is robust.

5. Add a more explicit discussion of the EM failure modes. The paper notes that Caps-CAM is weaker on EM in some settings, including MNIST and MSTAR. A deeper analysis of why the relevance construction is less reliable there would make the paper more informative and more credible.

**These are suggestions, but would strengthen the work:**

1. Expand the evaluation beyond perturbation-style metrics if possible. Average % Drop and AOPC are useful, but both are perturbation-based and can be sensitive to the evaluation protocol. Even one additional complementary criterion would help triangulate explanation quality.

---

> ### Author Response · Authors · 2026-05-02
> **Response to Reviewer 77p8**
>
> We thank the reviewer for the careful reading of our manuscript and for the constructive feedback. Below, we address each comment separately and describe the changes that will be incorporated in the revised version. Please note that all the changes can be found in blue in the revised manuscript.
> 1. We thank the reviewer for this constructive feedback(comment 1). We agree that including direct comparisons with CapsNet-specific explanation methods would strengthen the experimental validation. However, to the best of our knowledge, and as discussed in Section 2, existing explanation methods for CapsNets fall into two categories: Adaptations of CNN-based explanation methods (e.g., gradient-based methods, Grad-CAM variants), and Routing-based methods, which rely explicitly on capsule routing mechanisms to derive explanations.
> - Regarding the first category, our experimental evaluation already includes widely used and representative methods (Grad-CAM, Grad-CAM++, and Score-CAM), which have been adapted to Capsule Network architectures.
> For the second category, routing-based explanation methods (e.g., [1–2]) produce outputs in a fundamentally different form, such as routing coefficients, capsule contribution scores, or activation paths between capsules, rather than spatial attributions over the input. - In contrast, the proposed method generates explanation heatmaps defined in the input space, where each spatial location is assigned an importance value. Therefore, as these two categories produce fundamentally different forms of outputs, they are not directly comparable to our method. This difference in the form and objective of the explanations explains why such methods are not included in our experimental evaluation.
> - We acknowledge that this distinction was not sufficiently emphasized in the original manuscript. To address the reviewer’s concern, we revised Sec. 2 to clearly explain the comparability limitations of existing methods.
>
> 2. We thank the reviewer for this comment(comment 2). We agree that claims such as "general method", "superiority", and "reduced computational overhead" require more comprehensive and outspoken evidence.
> - In the revised manuscript, we have refined the wording to ensure all claims are appropriately scoped and supported by the presented evidence. Specifically, we replaced "superiority" with wording that reflects the scope of our empirical evaluation, and clarified that "general method" refers to applicability across feed-forward CapsNet variants rather than CapsNet architectures in general.
> - Regarding computational overhead, we have softened this claim to note that restricting computations to the most relevant capsules avoids redundant processing that would otherwise be required, without asserting a quantified reduction. We acknowledge that a formal runtime complexity comparison would better support this claim and agree it is a valuable direction for future work.
>
> 3. We thank the reviewer for this important suggestion(comment 3). In the revised manuscript, we have included a step-by-step pseudo-code describing the Caps-CAM procedure (Algorithm 1, Section 3.3). We have also revised the corresponding section to further clarify the method and address this concern. In addition, we will make the code publicly available upon acceptance to enhance the reproducibility of our method. We believe that these additions improve both the clarity of the method and the readability of the paper.
>
> 4. To address this concern(comment 4), we revised Sec 4 in the main manuscript and moved some of the corresponding experiments from the supplementary material to the main paper (Table.3 and Fig.10) . Specifically, we included qualitative comparisons of heatmaps obtained using different numbers of top-ranked capsules (e.g., top-2, top-4, top-6, and all capsules), as well as a discussion for the quantitative analysis.
> - This change allows us to present the results together with the related discussion, which makes the explanation more explicit and easier to follow. Specifically, we will summarize the results of varying Z (including top-2, top-4, and higher values) and explicitly highlight the observed trade-offs between noise suppression and information preservation.
> We believe that including these experiments in the main text will improve the clarity of the presentation and better support the claims made in this section.
>
> 5. We thank the reviewer for this suggestion (comment 5). In the revised manuscript (Sec. 4.1), we expand the discussion (MNIST) to clarify that the reduced reliability under EM routing could come from its clustering-based aggregation, which produces smoother but less selective relevance signals. As a result, the generated heatmaps may highlight relevant local features, while the final class assignment does not fully align with these features.

---

### Review · Reviewer_Nct2 · 2026-05-11

**Summary Of Contributions:**

The paper have made the following claims:
- The literature lacks the explainable solutions for Capsule Networks.
- Caps-CAM, a method combining Score-CAM and gradient information, is proposed to address this limitation.
- The experiments show the effectiveness of the proposed method on both classic and real-world datasets.

**Audience:**

No

**Audience Explanation:**

It's really hard to assess if the audience in TMLR will be interested in the findings of this paper. The motivation, positioning and the contributions are all unclear in the current manuscripts. Detailed comments are provided below.

**Broader Impact Concerns:**

As Capsule Network is a general framework that can be applied to tasks in various tasks including medical or person-identity domain, the reviewer suggests that the submission should still add a Broader Impact Statement section to discuss on the potential impact it may cause.

**Claims And Evidence:**

No

**Claims Explanation:**

Unfortunately no. The claims and contributions are unclear, the clarity of methodology description is poor, and the evaluations to support the claims are insufficient. Detailed feedback is provided in "Requested Changes" section.

**Requested Changes:**

- It will let the reviewer to better assess this work's impact if the authors can clarify its position and contribution. Does it want to propose a SOTA Capsule Network method or an explainability analysis on Capsule Network methods? Then, according to the clarified intention above, provide more comprehensive evaluation to better reflect this. The current evaluations are quite insufficient, it did not contain any statistical analysis or in-depth discussion.

- Revise and proofreading the submission thoroughly. The narrative and clarity of current manuscript requires significant improvement. The reviewer cannot understand the intention, methodology design and evaluation after reading the contexts multiple times.

More specifically,
1. What's the main challenges and limitations of current Capsule Network literature?
2. Why combine Score-CAM and gradients again? One main claim in Score-CAM is that the gradient is not reliable and has the gradient vanishing or exploding issue, why now it works? From Table 2, the Grad-CAM series method perform poorly, why selecting gradients by the top-2 activation maps doesn't have this issue?
3. Leverage the notation rigorously. In Sec 3.1 texts, the output of $F(x_i)$ function is the score $S_i$, however in Eq .1, it becomes activations $A$? how does that work? Moreover, for the sentence "the activations $A^k_l \in R^{1 \times Z}$ (with $Z$ being the dimensionality of the activations output) are transposed to rearrange their dimensions, yielding $A^{k,⊺}_{l} (x_i) \in R^{1×Z×K×H×W}$", how to get three more dimensions by transposing a matrix? There are many samples not mentioned. Please review the whole manuscript carefully.
4. The paper is not self-contained. Abstract, Introduction, and Related Work sections are highly overlapped with duplicate sentences, the section 3.1 did not introduce Score-CAM clearly. The reviewer needs to read the Score-CAM paper itself to understand how it works.
- Any claim made without the measurable metrics is fragile. Providing qualitative results are appreciated, however, the connection between the proposed method and the performance improvement should be examined more carefully. The current quantitive results are extremely insufficient. All experiments are conducted on small-scale datasets, without the discussion on how to scale up to larger dataset like ImageNet, or images from different domains, etc, which significantly limits the potential impact the work may bring.

---

> ### Author Response · Authors · 2026-05-24
> **Response to Reviewer Nct2**
>
> We thank the reviewer for the careful reading of our manuscript and for the constructive feedback. Below we address each comment separately and describe the changes that have been incorporated in the revised version. Please note that all the changes can be found in blue in the revised manuscript.
>
> •	It will let the reviewer to better assess this work's impact if the authors can clarify its position and contribution. Does it want to propose a SOTA Capsule Network method or an explainability analysis on Capsule Network methods? Then, according to the clarified intention above, provide more comprehensive evaluation to better reflect this. The current evaluations are quite insufficient, it did not contain any statistical analysis or in-depth discussion.
> 1. We thank the reviewer for this comment. We clarify that the goal of this work is not to propose a new Capsule Network architecture, but rather to introduce Caps-CAM, a novel post-hoc explainability method specifically designed for justifying the predictions of  existing feed-forward CapsNets without modifying their architecture , routing mechanisms, or training procedure.
> 2. To address this concern, we revised the manuscript and clarified this point in both the Introduction and Section 3.

---

> > ### Author Response · Authors · 2026-05-24
> > **Response to Reviewer Nct2**
> >
> > •	Revise and proofreading the submission thoroughly. The narrative and clarity of current manuscript requires significant improvement. The reviewer cannot understand the intention, methodology design and evaluation after reading the contexts multiple times.
> >
> > •	More specifically,
> >
> > * What's the main challenges and limitations of current Capsule Network literature?
> >
> > 1. We thank the reviewer for this comment. To clarify this point, we revised the Related Work (Section 2) section to explicitly state that the primary objective of our work is to explain the predictions made by CapsNets through a post-hoc analysis framework.
> > 2. Therefore, the challenges and limitations discussed in the manuscript are specifically related to the explainability of CapsNet architectures, rather than to improving CapsNet architectures themselves. In particular, we now emphasize the limited availability of explanation methods specifically designed for CapsNets in the existing literature, which motivates the need for the proposed Caps-CAM method.

---

> ### Author Response · Authors · 2026-05-24
>
> * Why combine Score-CAM and gradients again? One main claim in Score-CAM is that the gradient is not reliable and has the gradient vanishing or exploding issue, why now it works? From Table 2, the Grad-CAM series method perform poorly, why selecting gradients by the top-2 activation maps doesn't have this issue?
>
> 1. We thank the reviewer for this important question regarding the role of gradients in Caps-CAM.
> 2. We would like to clarify that Caps-CAM does not rely only on gradients for generating the final explanation maps. Instead, gradients are used only during the selection stage to identify the most informative capsule dimensions associated with the predicted class. The final explanation is then generated by combining these selected top capsule dimensions and their activation maps using score-based weighting.
> 3. In addition, during training, we monitored the optimization and loss curves and avoided excessive convergence to poor local minima or saturated regions. As a result, the gradient information extracted from the trained models remained informative and stable for the proposed selection process. Therefore, it helps reduce the limitations commonly associated with purely gradient-based explanations.
> 4. In the worst-case scenario, when the extracted gradients become very small but are not exactly zero, the selection of capsule dimensions may become less reliable, which can lead to less accurate heatmaps. In such cases, the generated explanations may highlight irrelevant features and therefore become less faithful to the model prediction. However, unlike Grad-CAM, the gradients in Caps-CAM are not directly used to generate the heatmap itself. Instead, they are only employed during the selection stage of the activation maps, while the final heatmap is produced from the gradients and activation maps themselves.
> 5. To address this concern, we added a paragraph by the end of Section 3.3 explaining the role of using gradients in Caps-CAM.
> 6. To complement the answer, our method outperform GradCAM by only selecting top capsules since our method does not employ the gradients as Grad-CAM does as explained above. Furthermore, Compared to other methods Caps-CAM clearly outperforms Grad-CAM and Grad-CAM++ (Table 2) . This improvement stems from two key factors. First, the masking of the input using relevance activation maps ensures that only the most informative regions of the input contribute to the explanation, which reduces noise from irrelevant regions. Second, the use of calculated confidence scores prioritizes capsules that are most predictive of the output, further emphasizing important regions. Together, these mechanisms enhance the reliability of the visual heatmaps, which explains the improved performance(section 4.2).

---

> ### Author Response · Authors · 2026-05-24
> **Response to Reviewer Nct2**
>
> * Leverage the notation rigorously. In Sec 3.1 texts, the output of  function is the score , however in Eq .1, it becomes activations ? how does that work?
>
> 1. 	We thank the reviewer for this important remark. We acknowledge that we previously misused the notation F(⋅)by using it to represent both the class scores and the intermediate activations. To address this issue, we have revised the manuscript and clarified the correct notation (Eq .1), explicitly distinguishing between the extraction of class scores and the extraction of intermediate activations.
> 2. Let $F(\cdot)$ denote the complete CapsNet model, which can be decomposed into a backbone feature extractor $b(\cdot)$ and a classification head. Given an input image $x_i$, the backbone first produces intermediate representations $A_i = b(x_i)$, which correspond to the capsule activations at a target layer. These activations are then passed to the classifier to produce the final class score vector $S_i = F(b(x_i))$, where $S_i \in \mathbb{R}^{C}$ and $C$ denotes the number of classes. Each element $S_i(c)$ represents the prediction score for class $c$. Therefore, $A_i$ and $S_i$ correspond to different stages of the same CapsNet architecture: $A_i$ is used to construct the visual explanation maps, while $S_i$ is used to determine the predicted class.
>
> 3. To compute the capsule relevance, gradients are obtained with respect to the predicted class score. Specifically, after determining the predicted class
>
> $\hat{y}_i = \arg\max_{c} S_i(c)$
>
> the gradients are calculated as
>
> $$
> G_l^k(x_i,\hat{y}_i) = \frac{\partial S_i(\hat{y}_i)}{\partial A_l^k(x_i)}.
> $$

---

> ### Author Response · Authors · 2026-05-24
> **Response to Reviewer Nct2**
>
> * Moreover, for the sentence "the activations $A^{k}P{l} \in R^{1 \times Z}ZA^{k,⊺}{l} (xi) \in R^{1×Z×K×H×W}$", how to get three more dimensions by transposing a matrix? There are many samples not mentioned. Please review the whole manuscript carefully.
>
> 1. We thank the reviewer for this important remark. We would like to clarify that the tensor A_l^k (x_i)corresponds to the actual output representation of the CapsNet layer, where capsule activations are already structured along multiple axes (including capsule type, instantiation parameters, and spatial dimensions depending on the architecture). Therefore, the representation is not obtained by increasing dimensionality through any transformation or operation; instead, it is an intermediate output of the network.
>
> 2. The reshaping and permutation steps applied in our method only reorganize this existing tensor structure to explicitly separate the capsule dimension from spatial dimensions, enabling capsule-wise processing. In particular, the operation simply reorders and flattens existing axes (e.g., grouping capsule instantiation parameters Z and spatial dimensions H×W) without introducing any additional computation or new features. Consequently, no new samples or hidden dimensions are created; the transformation is purely structural and preserves all original information produced by the CapsNet layer.
>
> 3. We have revised the manuscript to make this tensor structure and reshaping operation clearer in order to avoid any ambiguity. In addition, we will make the code publicly available upon acceptance to enhance the reproducibility of our method. We believe that these additions improve both the clarity of the method and the overall readability of the paper.

---

> ### Author Response · Authors · 2026-05-24
> **Response to Reviewer Nct2**
>
> * The paper is not self-contained. Abstract, Introduction, and Related Work sections are highly overlapped with duplicate sentences, the section 3.1 did not introduce Score-CAM clearly. The reviewer needs to read the Score-CAM paper itself to understand how it works.
>
> 1. We thank the reviewer for this valuable comment. The goal of Sec. 3.1 was not to reproduce the complete Score-CAM methodology, but rather to summarize the key steps that are directly relevant to understanding the proposed Caps-CAM adaptation. Since Score-CAM is an established explainability method, we focused primarily on the modifications required to extend it to CapsNets rather than restating the entire original framework.
>
> * Any claim made without the measurable metrics is fragile. Providing qualitative results are appreciated, however, the connection between the proposed method and the performance improvement should be examined more carefully. The current quantitative results are extremely insufficient. All experiments are conducted on small-scale datasets, without the discussion on how to scale up to larger dataset like ImageNet, or images from different domains, etc, which significantly limits the potential impact the work may bring.
>
> 1. We agree that quantitative validation is essential for establishing the reliability of explainability methods. The evaluation metrics adopted in this work are widely used and well-established in the explainability literature for evaluating the quality and faithfulness of visual explanations. To further strengthen the empirical analysis, we expanded the quantitative evaluation by introducing an additional confidence increase analysis, which measures the effect of the highlighted regions on the model’s prediction confidence. The corresponding results have been added to Table 3, together with an extended discussion in Section 4.2.
> 2. Across all architectures and datasets, Caps-CAM consistently achieves competitive or superior performance in terms of average % increase in confidence, particularly for the DR and DR-Conv models, where it ranks best overall. While a few isolated cases (e.g., DR-Tra on MNIST and EM on CIFAR-10) show slightly lower gains compared to alternatives, Caps-CAM remains generally strong across settings. Together with the average % drop results, these findings indicate that the regions identified by Caps-CAM are both sufficient to sustain predictions and highly influential when emphasized, reinforcing the method’s ability to reliably capture the most discriminative features used by CapsNet models.
>
> 3. Regarding the use of relatively small-scale datasets, we would like to clarify that the selected datasets are standard benchmark datasets that are used in CapsNets literature [1, 2, 3, 4, 5]. Their use enables fair comparison with prior works while maintaining consistency with the evaluation protocols commonly adopted in the literature. Moreover, CapsNets are known to involve substantially higher computational complexity and routing overhead compared to conventional CNNs, which explains why many existing CapsNet studies are evaluated on similar small benchmark datasets such as MNIST or SVHN.
>
> 4. To address the reviewer’s concern regarding scalability, we expanded the discussion in the revised manuscript to clarify the current limitations that prevent CapsNets from scaling efficiently to large-scale datasets, primarily due to the computational cost associated with dynamic routing mechanisms (Section 4)
>
> 5. CapsNets face significant challenges in scaling to large-scale datasets such as ImageNet. This limitation derives from several factors: the iterative routing-by-agreement mechanism introduces substantial computational cost that grows with the number of capsules and classes; large-scale datasets exhibit high variability in object pose, appearance, and background, which can destabilize part-whole agreement; and the absence of widely optimized and efficient implementations further exacerbates scalability constraints. Therefore, these factors make CapsNets more difficult to scale compared to more efficient architectures such as CNNs and Transformers.
>
>
> [1] S. Sabour, N. Frosst, G. E. Hinton, Dynamic routing between capsules, Advances in neural information processing systems 30 (2017).
>
> [2] J. Gu, Interpretable graph capsule networks for object recognition, in: Proceedings of the AAAI Conference on Artificial Intelligence, Vol. 35, 2021, pp. 1469–1477.
>
> [3] H. Ren, J. Su, H. Lu, Evaluating generalization ability of convolutional neural networks and capsule networks for image classification via top-2 classification, arXiv preprint arXiv:1901.10112 (2019).
>
> [4] X. Ning, W. Tian, W. Li, Y. Lu, S. Nie, L. Sun, Z. Chen, Bdars capsnet: Bi-directional attention routing sausage capsule network, IEEE Access 8 (2020) 59059–59068.
>
> [5] R, Riccardo, E Tartaglione, and M Grangetto. "Capsule networks do not need to model everything." Pattern Recognition (2025): 112119.

---

### Decision · Action_Editor_v98E · 2026-06-03

**Recommendation:** Accept with minor revision

**Additional Comments:**

Please make these changes to the manuscript as part of the minor revisions.

1) Tighten the related work section by including recent or key equivariance/invariance methods based on capsules (check ICCV2025, ECCV, NeurIPS, TMLR, etc., for suitable references)
2) The paper should avoid very broad claims, such as "faithful", and instead focus on the fact that this is a post-hoc visual explanation method evaluated on feed-forward image CapsNets
3) Add an explicit limitation on explanation evaluation. The saliency literature shows that visual plausibility can be misleading, and attribution evaluation is inherently difficult because ground-truth feature importance is often unavailable. Perturbation-style metrics are useful but can be sensitive to the evaluation protocol and distribution shift (see, for instance, this paper: https://link.springer.com/article/10.1007/s10994-024-06550-x?utm).
4) Clarify the gradient-selection issue. The paper should make it clear that Caps-CAM does not prove gradients are stable; it uses them as a relevance heuristic whose reliability is tested empirically.
5) Please double-check all notations and algorithms again for any remaining mismatches.
6) Make the baseline limitation explicit. The paper does not directly compare with routing-based CapsNet explanation methods. The “different output space” argument is fine, but it must remain a limitation rather than a feature of the method.

**Audience:**

Yes

**Audience Explanation:**

The topic of the paper is definitely within the broader topics TMLR welcomes, and it will appeal to several researchers in the evaluation/explainability domain, especially given that we need different perspectives and approaches across different architectures and methods. Therefore it is positive that capsules are being explored here, given that even recent papers show that in low data regime and without heavy pretraining, capsules remain very competitive, but are overshadowed by large pretrained and foundation models.

**Claims And Evidence:**

Yes

**Claims Explanation:**

The reviewers provided good overview of the strengths and weaknesses of the paper, and the authors provided good responses and improved the manuscript accordingly. Therefore, the claims and evidence meet the TMLR standard. I agree with the more positive-inclined reviewers about the narrow scope of the technical aspects of the paper, which proposes and evaluates a post-hoc visual explanation method for feed-forward image Capsule Networks, with empirical support against adapted CAM-style heatmap baselines across several capsule architectures and datasets. I think this is perfectly fine.

Similarly, the paper should not be read as a definitive explanation framework for all capsule architectures. The absence of more generic comparisons to routing-based explanation methods, the mixed EM-routing results, the reliance on perturbation/confidence-style metrics, and the use of gradients for capsule selection remain limitations. I have provided some revisions for the authors to make before the manuscript is ready for publication.

---

> ### Author Response · Authors · 2026-06-30
> **Response to the Action Editor**
>
> We thank the Action Editor for the positive assessment of our work and for the final minor suggestions. We have addressed the requested revisions as follows:
>
> 1. We added relevant work from top-tier venues (ICCV 2025, ECCV, NeurIPS, and TMLR) related to the current submission, including recent capsule-based methods on equivariance and invariance.
>
> 2. We revised the manuscript to avoid overly broad claims (e.g., "faithful") and consistently describe Caps-CAM as a post-hoc visual explanation method for feed-forward image Capsule Networks. Furthermore, we performed a final review to avoid overly strong statements throughout the manuscript.
>
> 3. We added an explicit discussion of the limitations of explanation evaluation, highlighting the challenges of attribution evaluation, the lack of ground-truth feature importance, and the limitations of perturbation-based metrics.
>
> 4. We clarified that Caps-CAM uses gradients solely as a relevance heuristic for capsule selection and does not claim that gradient-based relevance is inherently stable; rather, its usefulness is supported by our empirical evaluation.
>
> 5. We carefully reviewed the manuscript to resolve the remaining notation inconsistencies and algorithmic mismatches.
>
> 6. We explicitly state as a limitation that our work does not include comparisons with routing-based CapsNet explanation methods, while explaining the differences in output space and problem setting, and we identify this as an important direction for future work.